# Palmitic but Not Oleic Acid Induces Pro-Inflammatory Dysfunction of Human Endothelial Cells from Different Vascular Beds In Vitro

**DOI:** 10.3390/ijms262412148

**Published:** 2025-12-17

**Authors:** Daria Shishkova, Victoria Markova, Yulia Yurieva, Alexey Frolov, Anastasia Lazebnaya, Maxim Sinitsky, Anna Sinitskaya, Vera Matveeva, Evgenia Torgunakova, Alexander Stepanov, Anna Malashicheva, Asker Khapchaev, Nikita Podkuychenko, Alexander Vorotnikov, Vladimir Shirinsky, Anton Kutikhin

**Affiliations:** 1Department of Experimental Medicine, Research Institute for Complex Issues of Cardiovascular Diseases, 6 Barbarash Boulevard, 650002 Kemerovo, Russia; shidk@kemcardio.ru (D.S.); markve@kemcardio.ru (V.M.); markyo@kemcardio.ru (Y.Y.); frolav@kemcardio.ru (A.F.); lazeai@kemcardio.ru (A.L.); sinimu@kemcardio.ru (M.S.); cepoav@kemcardio.ru (A.S.); matvvg@kemcardio.ru (V.M.); torgea@kemcardio.ru (E.T.); stepad@kemcardio.ru (A.S.); 2Laboratory of Regenerative Biomedicine, Institute of Cytology of the RAS, 4 Tikhoretskiy Prospekt, 194064 St. Petersburg, Russia; malashicheva@incras.ru; 3Laboratory for Cell Motility, Institute of Experimental Cardiology, National Medical Research Centre of Cardiology Named After Academician E.I. Chazov, 15A Academician Chazov Street, 121552 Moscow, Russia; aykhapchaev@cardio.ru (A.K.); nra.fox@gmail.com (N.P.); a.vorotnikov@icloud.com (A.V.); shirinsky@gmail.com (V.S.)

**Keywords:** endothelial cells, endothelial dysfunction, atherosclerosis, vascular inflammation, aortic valve, saphenous vein, internal thoracic artery, microvascular network, palmitic acid, oleic acid

## Abstract

Palmitic acid (PA) is the most common dietary saturated fatty acid, and is abundant in palm and cottonseed oil, butter, and cheese, whereas oleic acid (OA) is a monounsaturated omega-9 fatty acid found in olive oil. The differences in the cytotoxic and pro-inflammatory effects of PA and OA across endothelial cells (ECs) isolated from different vascular beds have not been investigated in detail. Here, we incubated primary human aortic valve (HAVEC), saphenous vein (HSaVEC), internal thoracic artery (HITAEC), and microvascular (HMVEC) ECs with albumin-bound PA or OA for 24 h and found that PA induced a considerable cytotoxic response, accompanied by an elevated expression of the genes encoding cell adhesion molecules (*VCAM1*, *ICAM1*, *SELE*, and *SELP*) and pro-inflammatory cytokines (*MIF*, *PTX3*, *CSF2*, *CSF3*, *IL1A*, *IL6*, *CCL2*, *CCL5*, *CCL20*, *CSF2*, *CSF3*, *CXCL1*, *CXCL2*, *CXCL3*, *CXCL5*, *CXCL6*, *CXCL8*, and *CXCL10*), followed by an increased release of interleukin-6 and interleukin-8. HAVEC and HSaVEC were more susceptible to PA, whereas OA had mild-to-moderate cytotoxic effects on HAVEC and HMVEC but did not induce generalized EC activation. Compared with other EC types, HITAEC was the most resistant to PA and OA treatment. Collectively, these results indicate considerable heterogeneity across the ECs of distinct origin in response to PA.

## 1. Introduction

Fatty acids are critical components of the human diet, serving as essential sources of energy and being indispensable components of cell membranes and adipose tissue [1,2]. Among the numerous fatty acids, palmitic acid (PA) and oleic acid (OA) are highly prevalent in the human diet and are notable for different chemical structures, dietary sources, and health implications [1,2]. PA is the most common saturated fatty acid in the human body, abundantly found in animal and plant sources (especially in palm oil, butter, cheese, and lard), whereas OA is the most common monounsaturated fatty acid in the human diet, and is particularly plentiful in olive oil, avocados, nuts, and seeds [1,2,3,4]. An increased consumption of PA adversely affects vascular health, correlating with elevated low-density lipoprotein cholesterol [4,5,6], insulin resistance [7,8,9,10,11,12], and pro-inflammatory states [11,12,13,14,15]. A lipid-rich Western-style diet, which is characterized by a high intake of processed foods, red meat, and dairy products with a significant amount of PA, has been linked to the augmenting rates of arterial hypertension and atherosclerotic vascular disease in the high-income and upper–middle-income countries [16,17,18,19,20,21]. On the contrary, dietary replacement of saturated fats with OA-rich oils, which exert anti-inflammatory action [22,23,24,25,26], is associated with reduced low-to-high density lipoprotein cholesterol ratio [27,28,29,30], better insulin sensitivity [31,32], and lower blood pressure [33,34,35]. Likewise, the Mediterranean-style diet, rich in OA due to the high contents of olive oil, is well known for its beneficial metabolic [36,37,38] and cardioprotective effects [36,37,38,39,40] and decreased risk of major adverse cardiovascular events [41,42,43,44,45]. Therefore, balancing the consumption of PA and OA is of crucial importance for successfully managing cardiovascular risk, particularly in genetically predisposed individuals [46,47,48,49,50].

The integrity and quiescent state of the endothelial monolayer have a pivotal significance for maintaining vascular homeostasis, since the endothelium provides multiple physiological cues to vascular smooth muscle cells located in the tunica media, as well as to adventitial fibroblasts and macrophages in the tunica adventitia and to perivascular adipose cells through the vasa vasorum network [51,52,53,54,55,56]. Endothelial dysfunction [57,58,59,60], triggered by metabolic disorders such as dyslipidemia, hyperglycemia, or uremia [61,62], contributes to a chronic low-grade inflammation [63,64], thus mediating the development of the inflammaging [65,66,67,68] and frailty syndrome [69,70]. Another consequence of pro-inflammatory endothelial dysfunction is pathological endothelial permeability, which promotes the retention of atherogenic lipoproteins in the tunica intima and the migration of monocytes which further transform into foam cells upon engulfing the lipid droplets [59,71,72,73]. Among the features of pro-inflammatory endothelial activation are the excessive secretion of interleukin (IL)-6, IL-8, monocyte chemoattractant protein 1/C-C motif ligand 2 (MCP-1/CCL2), regulated on activation, normal T-cell expressed and secreted (RANTES/CCL5), and macrophage inflammatory protein 3 alpha (MIP-3α/CCL20), and combined overexpression of the genes encoding pro-inflammatory cytokines, cell adhesion molecules (*VCAM1*, *ICAM1*, *SELE*, and *SELP*), and endothelial-to-mesenchymal transition (EndoMT) transcription factors (*SNAI1*, *SNAI2*, *TWIST1*, and *ZEB1*) [74,75,76,77,78]. The immediate upregulation of the latter transcripts reflects non-specific endothelial stress rather than the ongoing loss of endothelial markers and the acquisition of mesenchymal markers [75,76,77,78]. Pro-inflammatory stimuli also initiate the apoptosis, pyroptosis, and necroptosis of the most vulnerable endothelial cells (ECs), although laminar flow in the blood vessels generally ameliorates these sequelae [79,80,81,82,83]. Nevertheless, soluble EC markers are considered as sensitive and relatively specific markers of pro-inflammatory endothelial activation, which is typically observed in patients with moderate-to-severe COVID-19 [84,85,86,87,88], septic shock [89,90,91,92,93], and frailty syndrome [94,95]. The delineation of the widespread risk factors of endothelial dysfunction and the implementation of the corresponding lifestyle, dietary, or pharmacological interventions might improve vascular health in the population, leading to the better cardiovascular outcomes.

Previous reports indicated the detrimental effects of PA, as well as the neutral or protective role of OA, on EC function. Exposure to PA promotes reactive oxygen species production and oxidative stress [96,97,98,99,100,101], activates NLRP3 inflammasome [96,102], enhances the expression of pro-inflammatory cytokines (IL-1β, IL-6, IL-8, and IL-18) and cell adhesion molecules (ICAM-1) [96,102,103], accelerates lipid accumulation [102], reduces nitric oxide (NO) production [101,103], and impairs endothelial barrier integrity [103], ultimately leading to the pyroptosis [96,102] and apoptosis [100,101] of the ECs. However, most of the existing studies have been restricted to a single EC type and have not assessed the effects of PA on ECs from distinct vascular beds. To delineate common and EC type-specific effects of PA-induced endothelial dysfunction, here we prepared the conjugates of PA and OA with a fatty acid-free bovine serum albumin (PA-BSA and OA-BSA, respectively), and compared the cytotoxic and pro-inflammatory effects of PA-BSA and OA-BSA on primary human aortic valve endothelial cells (HAVEC), human saphenous vein endothelial cells (HSaVEC), human internal thoracic artery endothelial cells (HITAEC), and human adipose tissue-derived microvascular endothelial cells (HMVEC) under identical cell culture conditions. We found that PA-BSA exerted significant cytotoxicity and caused pro-inflammatory activation in all the indicated EC types, primarily HAVEC and HSaVEC, whereas OA-BSA had mild-to-moderate cytotoxic effects on HAVEC and HMVEC but did not induce generalized EC activation. Across the four EC types studied, HITAEC was the most resistant to PA-BSA exposure and was also not affected by OA-BSA. Taken together, these results indicate considerable heterogeneity across the ECs of distinct origin in response to PA-BSA.

## 2. Results

We first asked whether the treatment with control fatty acid-free BSA, PA-BSA (0.8 mmol/L), or OA-BSA (0.8 mmol/L) for 24 h induces cytotoxic effects in HAVEC, HSaVEC, HITAEC, or HMVEC. BSA was selected as a carrier protein for PA and OA to ensure their internalization by the ECs. The cytotoxicity of PA-BSA and OA-BSA was assessed by cell visualization and a microplate colorimetric assay evaluating the reduction in a water-soluble tetrazolium salt (WST-8) by the intracellular dehydrogenases into a water-soluble orange–yellow formazan compound with a maximum absorption at a 450 nm wavelength. Upon phase contrast microscopy, cytotoxic effects were defined as loss of the typical cobblestone morphology, EC detachment and rounding, and the presence of floating and fragmented cells, collectively indicating the disruption of endothelial monolayer integrity if supported by the reduction in cell proliferation and viability at the WST-8 assay.

Treatment with PA-BSA induced the demise and detachment of HAVEC (Figure 1), HSaVEC (Figure 2), HITAEC (Figure 3), and HMVEC (Figure 4), ultimately leading to the loss of the cobblestone appearance and impairment of the EC monolayer, whilst ECs incubated with OA-BSA resembled the control ECs at visual inspection (Figure 1, Figure 2, Figure 3 and Figure 4). The colorimetric WST-8 assay revealed that PA-BSA exhibited pronounced cytotoxicity in all EC types, resulting in a significant reduction in cell viability (Figure 1, Figure 2, Figure 3 and Figure 4). The most pronounced decrease in metabolic activity was observed in HAVEC (Figure 1) and HSaVEC (Figure 2), which appeared to be the most sensitive to PA-BSA treatment. The effects of OA-BSA were more variable and cell type-dependent, with HAVEC (Figure 1) and HMVEC (Figure 4) showing a mild-to-moderate decline in viability, whereas HSaVEC (Figure 2) and HITAEC (Figure 3) remained unaffected. These findings highlight the heterogeneity of the responses between distinct EC types, suggesting that the extent of fatty acid-induced cytotoxicity is largely determined by the EC origin.

We next aimed to explore the molecular consequences of PA-BSA and OA-BSA treatment of HAVEC, HSaVEC, HITAEC, and HMVEC. Reverse transcription–quantitative polymerase chain reaction (RT-qPCR) profiling revealed a ≥2.5-fold increase in the expression of all genes encoding cell adhesion molecules (*VCAM1*, *ICAM1*, *SELE*, and *SELP*), whereas OA-BSA did not cause such effects (Figure 5). Across these pro-inflammatory receptors, the *ICAM1* gene was the most upregulated (≈7.5-fold), whereas the *VCAM1*, *SELE*, and *SELP* genes exhibited lower (≈2.5-fold) overexpressions (Figure 5). The expression of all 16 genes encoding core endothelial pro-inflammatory cytokines (*MIF*, *PTX3*, *CSF2*, *CSF3*, *IL1A*, *IL6*, *CCL2*, *CCL5*, *CCL20*, *CSF2*, *CSF3*, *CXCL1*, *CXCL2*, *CXCL3*, *CXCL5*, *CXCL6*, *CXCL8*, and *CXCL10*) was significantly (≥2-fold) increased in PA-BSA-treated ECs, whilst none of them were upregulated in ECs incubated with OA-BSA (Figure 5). Among the indicated genes, the highest overexpression was detected for the *IL6* (≈18-fold), *CXCL8* (≈16-fold), *CCL20* (≈11-fold), *CXCL2* (≈9-fold), *PTX3* (≈9-fold), *IL1A* (≈9-fold), *CCL5* (≈9-fold), and *CSF2* (≈8-fold) genes (Figure 5). The lowest overexpression was shown by *CCL2* gene (1.9-fold, Figure 5). The expression of the genes encoding EndoMT transcription factors (*SNAI1*, *SNAI2*, *TWIST1*, and *ZEB1*), which are generally upregulated at endothelial dysfunction, also showed ≥2-fold increase after PA-BSA treatment, in particular the *TWIST1* (≈eight-fold increase) and *SNAI1* genes (≈four-fold increase, Figure 5). The *TWIST1* and *ZEB1* genes also demonstrated a ≥two-fold increase following the incubation of ECs with OA-BSA (Figure 5).

We then evaluated the heterogeneity of endothelial transcriptional response to PA-BSA and OA-BSA. The treatment of HAVEC with PA-BSA led to the significant increase in the expression of genes encoding 13 pro-inflammatory cytokines (*MIF*, *PTX3*, *CSF2*, *IL1A*, *IL6*, *CCL2*, *CCL5*, *CXCL1*, *CXCL2*, *CXCL3*, *CXCL5*, *CXCL6*, and *CXCL8*), whilst only *MIF* gene expression was elevated upon the incubation with OA-BSA (Figure 6). The expression of the genes encoding cell adhesion molecules (*VCAM1*, *ICAM1*, *SELE*, and *SELP*) did not show statistically significant alterations, though the *VCAM1*, *ICAM1*, and *SELE* genes were overexpressed in HAVEC after PA-BSA treatment (Figure 6). The treatment of HAVEC with either PA-BSA or OA-BSA induced the expression of the genes encoding EndoMT transcription factors (*SNAI1*, *SNAI2*, *TWIST1*, and *ZEB1*, Figure 6), indicating the development of endothelial stress in keeping with the reduced viability of HAVEC in the WST-8 assay (Figure 1).

Similarly to HAVEC the treatment of HSaVEC with PA-BSA elevated the expression of 13 genes encoding pro-inflammatory cytokines (*MIF*, *PTX3*, *CSF2*, *CSF3*, *IL1A*, *IL6*, *CCL2*, *CCL20*, *CXCL1*, *CXCL3*, *CXCL5*, *CXCL6*, and *CXCL8*), whereas OA-BSA stimulated the expression of only four genes in this category (*PTX3*, *CCL20*, *CXCL3*, and *CXCL10*, Figure 7). The expression of the genes encoding cell adhesion molecules in HSaVEC also tended to increase upon PA-BSA exposure, albeit only *ICAM1* and *SELP* genes were significantly overexpressed (Figure 7). Notably, PA-BSA induced the expression of the genes encoding EndoMT transcription factors *(SNAI1* and *ZEB1*, Figure 7). The treatment of HSaVEC with OA-BSA did not reveal a consistent increase in the expression of genes within these categories (Figure 7).

In concert with the results of the WST-8 assay, the incubation of HITAEC with PA-BSA led to a lower endothelial response as compared with HAVEC and HSaVEC, resulting in the significant increase in only one gene (*CCL5*), although the expression of eleven cytokine genes (*PTX3*, *CSF2*, *CSF3*, *IL1A*, *IL6*, *CCL20*, *CXCL1*, *CXCL2*, *CXCL3*, *CXCL5*, and *CXCL8*) also tended to be augmented (Figure 8). OA-BSA did not cause such alterations in the cytokine gene expression in HITAEC (Figure 8). Along similar lines, the expression of the genes encoding cell adhesion molecules (*VCAM1*, *ICAM1*, *SELE*, and *SELP*) and EndoMT transcription factors (*SNAI1*, *SNAI2*, *TWIST1*, and *ZEB1*) in HITAEC showed a notable but statistically insignificant increase upon PA-BSA but not upon OA-BSA treatment (Figure 8).

The treatment of HMVEC with PA-BSA led to the increased expression of eight of the sixteen genes encoding pro-inflammatory cytokines (*CSF2*, *IL6*, *CCL20*, *CXCL1*, *CXCL2*, *CXCL3*, *CXCL5*, and *CXCL8*); however, the expression of other genes (*MIF*, *PTX3*, *CSF3*, *IL1A*, *CCL2*, *CCL5*, *CXCL6*, and *CXCL10*) also showed insignificant increment (Figure 9). The expression of the *VCAM1*, *ICAM1*, and *SELP* genes tended to be elevated after PA-BSA treatment, although only the *ICAM1* gene demonstrated a statistically significant increase (Figure 9). The expression of the genes encoding EndoMT transcription factors (SNAI1, SNAI2, TWIST1, and ZEB1) remained unchanged upon PA-BSA exposure (Figure 9). None of the indicated genes were stimulated by OA-BSA (Figure 9). In summary, the results of RT-qPCR profiling were in concord with those obtained by phase contrast microscopy and the WST-8 cell proliferation and viability assay, showing the higher susceptibility of HAVEC and HSaVEC to PA-BSA.

For the verification of the RT-qPCR results, we conducted an enzyme-linked immunosorbent assay (ELISA) to measure the cytokines with which genes demonstrated the highest average (IL-6 and IL-8) and lowest average (MCP-1/CCL2) overexpression after the incubation of ECs with PA-BSA (Figure 5). The PA-BSA treatment induced the release of IL-6 and IL-8, whilst the level of MCP-1/CCL2 remained unchanged (Figure 10), confirming the results of RT-qPCR profiling (Figure 5). Exposure to OA-BSA did not alter the production of IL-6, IL-8, and MCP-1/CCL2 (Figure 10). Across the studied EC types, the increase in IL-6 after PA-BSA treatment was the highest in HAVEC, HSaVEC, and HMVEC (Figure 10), confirming the greater pro-inflammatory effects of PA-BSA on HAVEC and HSaVEC in comparison with HITAEC.

In order to perform a systematic analysis of EC type-specific responses, we carried out a dot blot profiling of 109 pro-inflammatory cytokines and chemokines in the cell culture supernatant withdrawn from HAVEC, HSaVEC, HITAEC, and HMVEC treated with OA-BSA or PA-BSA (Figure 11, Figure 12, Figure 13 and Figure 14, Table 1). Similarly to RT-qPCR results, dot blotting measurements identified that HSaVEC showed the most pronounced response to PA-BSA (27 upregulated cytokines, Figure 11) in comparison with HMVEC (12 overexpressed cytokines, Figure 12), HITAEC (10 overrepresented cytokines, Figure 13), and HAVEC (7 upregulated cytokines, Figure 14). HSaVEC exposed to PA-BSA exhibited a greater pro-inflammatory response than that treated with OA-BSA (27 and 19 upregulated cytokines, respectively, Figure 11), whereas HMVEC was slightly more sensitive to OA-BSA than to PA-BSA (15 and 12 overexpressed cytokines, respectively, Figure 12). The indicated susceptibility of HMVEC to the OA-BSA treatment at dot blot profiling (Figure 12) also corresponded to the findings revealed at RT-qPCR analysis. In concert with RT-qPCR and ELISA, dot blot profiling demonstrated the notable resistance of HITAEC to both PA-BSA and OA-BSA (ten overexpressed cytokines after both treatments, Figure 13).

Among the cytokines overrepresented in the cell culture supernatant from HSaVEC after the PA-BSA treatment were IL-6, IL-8, MCP-1/CCL2, pentraxin-3, macrophage migration inhibitory factor (MIF), macrophage inflammatory protein-3α (MIP-3α), granulocyte–macrophage colony-stimulating factor (GM-CSF), as well as chemokines CXCL5 and CXCL11 (Figure 11). The exposure of HSaVEC to PA-BSA stimulated the production of pro-angiogenic molecules—hepatocyte growth factor, platelet-derived growth factor, angiogenin, trefoil factor 3, insulin-like growth factor binding protein-2, angiopoietin-2, and growth differentiation factor 15—and promoted the release of soluble endothelial cell receptors endoglin/CD105, PECAM1/CD31, VCAM1/CD106, and basigin/CD147, together suggesting a development of endothelial dysfunction (Figure 11). In contrast, the pro-inflammatory and pro-angiogenic response of HITAEC to PA-BSA was limited to the increased release of interleukin-1α, granulocyte colony-stimulating factor (G-CSF), GM-CSF, angiopoietin-2, and growth differentiation factor 15 (Figure 13). Collectively, the results of dot blot profiling corroborated those of RT-qPCR, indicating the notable heterogeneity regarding the response of different EC types to PA-BSA and OA-BSA.

As pro-inflammatory endothelial activation is intimately connected with oxidative stress, we further tested whether PA-BSA or OA-BSA induce the generation of reactive oxygen species (ROS). Treatment with PA-BSA significantly increased ROS formation, and ROS levels were from 2- to 3-fold higher in PA-BSA-treated than in BSA- or OA-BSA-treated cells (Figure 15). Exposure to OA-BSA led to the minor and insignificant ROS increase in OA-BSA-treated cells as compared to control (i.e., BSA-treated) cells (Figure 15). We then asked whether the treatment with antioxidant enzymes—i.e., superoxide dismutase (SOD) and catalase (CAT)—enables the rescue of ECs from PA-induced death. The co-incubation of HAVEC, HSaVEC, HITAEC, and HMVEC with SOD (250 U/mL), and CAT (500 U/mL) for 24 h partially prevented loss of their viability during the PA-BSA treatment (Figure 16). HAVEC and HSaVEC were particularly sensitive to PA-BSA, and the rescuing effect of SOD and CAT was highest in these EC types (≈73% for HAVEC and 37% for HSaVEC as compared to 23% in HITAEC and 28% in HMVEC after the PA-BSA treatment and combined addition of SOD and CAT to the cell culture medium, Figure 16). As compared with HSaVEC and HITAEC, HAVEC and HMVEC were slightly yet insignificantly affected by OA-BSA (Figure 16). The addition of SOD and CAT did not improve the viability of OA-BSA-treated ECs that concurred with the absence of ROS generation upon the exposure to OA-BSA (Figure 16). Hence, PA-BSA-induced endothelial dysfunction was accompanied by elevated ROS generation and was ameliorated by treatment with antioxidant enzymes (SOD and CAT).

To further examine whether the antioxidants can rescue ECs from PA-induced death, we tested the effects of L-ascorbic acid 2-phosphate (L-AA2P, 50 µg/mL) and sodium selenite (NaSe, 7 µg/mL), two common antioxidant additives to EC culture medium, on HAVEC, HSaVEC, HITAEC, and HMVEC treated with PA-BSA or OA-BSA. The addition of L-AA2P significantly increased the viability in all EC types (Figure 17). Similarly to SOD and CAT, the L-AA2P rescuing effect was higher in HAVEC and HSaVEC, which were more affected by PA-BSA (≈33% increment in viability in HAVEC, ≈30% in HSaVEC, ≈11% in HITAEC, and ≈18% in HMVEC after the PA-BSA treatment and L-AA2P supplementation as compared to PA-BSA-treated ECs without L-AA2P, Figure 17). In keeping with these findings, the addition of NaSe significantly elevated the viability of ECs regardless of their specification, although with a lower efficiency than SOD/CAT and L-AA2P (≈21% increment in viability in HAVEC, ≈17% in HSaVEC, ≈11% in HITAEC, and ≈15% in HMVEC after the PA-BSA treatment and NaSe supplementation as compared to PA-BSA-treated ECs without NaSe, Figure 18). Collectively, these results indicated the considerable role of oxidative stress in defining PA-specific endothelial toxicity and suggested the efficiency of antioxidant interventions in mitigating these detrimental effects.

## 3. Discussion

Here we report the differential effects of PA-BSA and OA-BSA on primary valvular and vascular ECs. The exposure of HAVEC, HSaVEC, HITAEC, and HMVEC to PA-BSA caused significant cytotoxic effects, induced the expression of the genes encoding cell adhesion molecules (*VCAM1*, *ICAM1*, *SELE*, and *SELP*), pro-inflammatory cytokines (*MIF*, *PTX3*, *CSF2*, *CSF3*, *IL1A*, *IL6*, *CCL2*, *CCL5*, *CCL20*, *CSF2*, *CSF3*, *CXCL1*, *CXCL2*, *CXCL3*, *CXCL5*, *CXCL6*, *CXCL8*, and *CXCL10*), and EndoMT transcription factors (*SNAI1*, *SNAI2*, *TWIST1*, and *ZEB1*), and promoted the production of IL-6 and IL-8 into the cell culture milieu. Across the indicated EC types, PA-BSA induced the highest cytotoxicity and the strongest pro-inflammatory activation in HAVEC and HSaVEC. For instance, the treatment of HSaVEC with PA-BSA initiated a prominent release of pro-inflammatory cytokines, soluble EC receptors, and pro-angiogenic molecules into the cell culture supernatant. Although OA-BSA also showed mild-to-moderate cytotoxicity in HAVEC and HMVEC, it did not cause consistent activation in the ECs. Of all the EC types studied, HITAEC demonstrated the highest resistance to PA-BSA and did not respond to the OA-BSA treatment. The findings revealed upon the phase contrast microscopy, the colorimetric cell proliferation and viability assay (WST-8), RT-qPCR, ELISA, and dot blot profiling corresponded to each other, together suggesting considerable heterogeneity across the ECs from different vascular territories in response to PA.

Previous studies have highlighted the following hazardous effects of PA on ECs: (1) mitochondrial [104,105] and endoplasmic reticulum stress [104] resulting in the increased production of a reactive oxygen species [104,105,106,107,108,109,110,111] in a NADPH oxidase-dependent [107,110,111] and Ca^2+^-dependent [109] manner; (2) inhibition of cell proliferation [104], migration [105], and angiogenesis [109]; and (3) accelerated apoptosis [104,105,108], and ferroptosis [112]. The pathological effects of PA on endothelial metabolism are largely mediated by an endoplasmic reticulum stress [113] indicated by an increase in respective biomarkers (GRP78, CHOP, PERK, and ATF4) [114], impaired autophagy [115] and respiratory burst [116], and pyroptosis represents a major EC death subroutine after the exposure to PA [117]. Notably, the inhibition of endoplasmic reticulum stress ameliorated PA-induced endothelial dysfunction in diabetic mice [118], and the inhibition of lipolysis in the visceral adipose tissue mitigated the PA uptake by the ECs [119]. Recent studies have also shown the importance of epigenetic regulation in determining the endothelial toxicity of PA [120], which may be particularly important for senescent ECs which accumulated a significant number of epigenetic alterations [121]. Together, these findings underscore the role of endoplasmic stress and metabolic inflammation in PA-induced lipotoxicity and PA-mediated endothelial dysfunction [122,123,124].

The indicated pathological pattern was accompanied by an upregulation of the *IL6*, *CCL2*, *VCAM1*, *ICAM1*, and *SELE* genes [105,106,107,108], primarily via the TLR4 pathway and the activation of NF-κB transcription factor [107,108,125,126,127,128]. Among the pro-inflammatory molecules released into the extracellular milieu upon PA stimulation were IL-6 [105,107,108,126], IL-8 [126], MCP-1/CCL2 [105], and pentraxin-3 [129]. Taken together, these molecular alterations led to an increased adhesion of monocytes to ECs [105,106,130,131] via ICAM-1 [126,130] and E-selectin [132], induced EndoMT [128], and impaired insulin signaling [108,125,127,132]. In addition, PA induced the expression of endothelin-1, a potent vasoconstrictor [108], diminished the production of a key vasodilator NO [108,109,111,125,127,128], and reduced acetylcholine-mediated endothelium-dependent vasodilation [110] but not vascular smooth muscle cell-dependent vasodilation [127]. In concert with the detrimental effects of PA on ECs, epidemiological studies have shown that elevated serum PA is associated with an increased risk of coronary artery disease [133,134], stroke [134], and cardiovascular death [135,136]. Our findings are consistent with these reports, as we also observed the enhanced transcription of multiple pro-inflammatory genes and the elevated secretion of IL-6 and IL-8 in ECs following PA treatment. However, a detailed investigation of specific signaling pathways involved in the EC response to PA requires further functional experiments (e.g., inhibition or rescue studies) in order to establish the causal conclusions.

In vitro experiments demonstrated a cell- and fatty acid-specific pattern of lipid deposition in liver sinusoidal ECs and hepatocytes, showing considerable chemical differences in PA- and OA-generated lipid droplets [137]. PA significantly inhibited the proliferation of ECs and caused an increase in caspase-3 expression [138]. Vice versa, OA slightly accelerated cell growth, although also leading to a mild elevation of caspase-3 [138]. The incubation of TNFα-stimulated ECs with palmitoleic acid and OA did not trigger an increase in IL-6, IL-8, and MCP-1/CCL2, though exposure to PA led to an elevation of these cytokines in the cell culture supernatant [139]. Similarly to palmitoleic acid, the combination of PA and OA abrogated the adverse effects regardless of the EC type [140]. Furthermore, incubation with oleate even reduced VCAM-1 [141] and ICAM-1 expression [142] and NF-κB activation [141,142], and OA mitigated the hazardous effects of other endothelial dysfunction triggers [141]. Whilst PA induced insulin resistance, OA protected against it, also reducing the release of MCP-1/CCL2 and augmenting eNOS biosynthesis [142]. OA-enriched extracellular vesicles contributed to cell viability and proliferation, whereas PA-modified extracellular vesicles promoted cell death via necroptosis [143]. Yet, other studies have found a mitochondrial dysfunction, a dose-dependent increase in mitochondrial-derived reactive oxygen species, a reduction in basal and insulin-dependent eNOS biosynthesis, an increase in endothelin-1 release, an NF-κB activation, and a moderate cytotoxicity after the exposure of ECs to OA [144,145,146]. The effects of OA might be cell-dependent, as oleate reduced energy production, impaired mitochondrial respiration and triggered cell death in liver sinusoidal ECs but not human umbilical vein ECs [140]. This corresponds to the findings of our study, in which OA showed a mild-to-moderate cytotoxicity in HAVEC and HMVEC but not HSaVEC and HITAEC, suggesting that the effects of OA significantly depend on the EC type.

After the digestion of fat in the small intestinal lumen, non-esterified PA and OA associate with bile salt micelles, are absorbed by enterocytes through passive diffusion and fatty acid transporters, and undergo re-esterification into triglycerides, which are further incorporated into chylomicrons [1,4,8,10,11]. Via the lymphatic system, chylomicrons then enter the bloodstream where endothelial lipoprotein lipase hydrolyzes triglycerides and release free PA and OA, which are further internalized by ECs and peripheral tissues by the abovementioned mechanisms, or remain in circulation as bound to albumin [1,4,8,10,11]. However, the pathophysiological and clinical consequences of PA excess need further investigation. A randomized controlled trial in this regard found a negative correlation between the OA-to-PA ratio (i.e., monounsaturated to saturated fatty acid ratio) and postprandial concentrations of tissue factor, fibrinogen, and plasminogen activator inhibitor 1, supporting arguments declaring the benefits of diets rich in olive oil [147]. In addition, the PA-enriched diet disrupts biological rhythms (i.e., the circadian clock), leading to the pathological alterations of sleep architecture and the physiological parameters during sleep even in healthy individuals [148]. Monocytes isolated from Western style-fed cynomolgus macaques were characterized by a pro-inflammatory transcriptional phenotype as compared with Mediterranean style-fed macaques [149]. Such a phenotype, in turn, was associated with increased anxiety and deteriorated social integration [149], corresponding to the abovementioned sleep disorders [148].

Even after the large systematic review and meta-analysis, it remains unclear whether the Mediterranean-style diet positively affects cardiovascular outcomes [150]. Yet, a shift from the lipid-rich Western-style to Mediterranean-style diet in order to perform a PA-to-OA replacement might have benefits for the patients with a high risk of myocardial infarction or ischemic stroke, preventing a repeated major adverse cardiovascular event. Such a PA-deficient and OA-enriched dietary pattern has the potential to delay the development of endothelial dysfunction and its pathophysiological (i.e., arterial hypertension, chronic low-grade inflammation, or atherosclerosis) and clinical (i.e., coronary artery disease, chronic brain ischemia, deep vein thrombosis, or frailty syndrome) consequences. Taking together our and previous findings, we suggest the advantage of PA restriction for maintaining endothelial homeostasis and vascular health. As the proper fatty acid intake is indispensable for sustainable metabolism and cell membrane synthesis, the partial replacement of PA with relatively harmless OA in the diet might be considered as a promising cardioprotective nutritional intervention. Further prospective studies or meta-analyses are required to test this hypothesis, which relies on the specific endothelial toxicity of PA as compared with OA.

Although we cannot completely exclude that inherited variation or specific disease-associated epigenetic or transcriptional adaptations could affect the cellular responses to PA-BSA and OA-BSA exposure, we performed all treatments (BSA, PA-BSA, and OA-BSA) in parallel for all EC types in order to minimize the patient-dependent variability. Although the extent of cytotoxic and pro-inflammatory effects differed between HAVEC, HSaVEC, HITAEC, and HMVEC (i.e., HAVEC and HSaVEC were more susceptible to PA-BSA whilst HAVEC and HMVEC were also affected by OA-BSA), PA-BSA consistently exerted considerable pathological effects across EC types. Such reproducibility suggests that the observed effects are primarily attributable to fatty acid exposure rather than to intrinsic genetic or disease-related background. Earlier studies of the pro-inflammatory effects of PA and the neutral or anti-inflammatory behavior of OA on the ECs from healthy donors and immortalized EC lines [137,138,139,140,141,142] provide additional evidence that these effects are specific to PA or OA rather than to the patient predisposition. We should also note that our study was focused on the investigation of cytotoxic and transcriptional response to PA and OA in the context of endothelial heterogeneity and did not interrogate the molecular mechanisms behind these effects. The advantages of our study include use of the following: distinct EC types (HAVEC, HSaVEC, HITAEC, and HMVEC), several donors per each EC type, and broad gene expression profiling. The study’s shortcomings include its limited proteomic profiling because of its use of fatty acid-free albumin, which restricts the application of ultra-high performance liquid chromatography–mass spectrometry but is necessary to carry PA and OA into ECs. Yet, the consistency of the results of phase contrast microscopy, colorimetric cell viability and proliferation assay, RT-qPCR, ELISA, and dot blotting suggests the considerable role of endothelial specification in defining the consequences of the exposure to PA or OA.

## 4. Materials and Methods

### 4.1. Preparation of PA-BSA and OA-BSA Conjugates

The PA-BSA and OA-BSA conjugates were prepared according to the previously published protocols [151,152] with modifications. Fatty acid-free BSA (≥98.0% purity, <0.01% fatty acids, PM-T1727, Biosera, Cholet, France) was dissolved to 20% in serum-free EndoLife medium (EL1, AppScience Products, Moscow, Russia) with the overnight rotation at 4 °C. The solution was then centrifuged for 40 min at 4500× *g* to remove any insoluble aggregates. The supernatant was sterilized by passing it through 0.22 μm syringe filters with a polyethersulfone membrane (Acrodisc, 4612, Cytiva, Marlborough, MA, USA). Sodium palmitate (≥98.5% purity, P9767, Sigma-Aldrich, Saint Louis, MO, USA) or sodium oleate (≥99% purity, O7501, Sigma-Aldrich, Saint Louis, MO, USA) was dissolved in 50% ethanol at 60 °C, transferred to the laminar hood, and added dropwise under constant stirring into fatty acid-free BSA solution, which was pre-warmed to 60 °C to achieve the final fatty acid concentration of 8.0 mmol/L and the 2.7 (mol:mol) ratio of fatty acid to BSA. Special care was taken to avoid any foaming of the solution and clouding due to palmitate precipitation. The control fatty acid-free BSA solution was made in the same fashion but using 50% ethanol without sodium palmitate or sodium oleate. Both solutions were kept for 20 min at 45 °C in a water bath with occasional stirring and then left for 5 days at 37 °C in a CO_2_ incubator (MCO-18AIC, PHCbi, Tokyo, Japan) to ensure complete conjugation of sodium palmitate and sodium oleate with a fatty acid-free BSA and the absence of residual fatty acid micelles. Finally, the PA-BSA and OA-BSA conjugates were sterilized by 0.22 µm filtration before their addition to cultured cells. Selected batches of conjugates were subjected to gas chromatography–mass spectrometry analysis with fatty acid standards to confirm that the added and measured amounts of fatty acids were equal.

### 4.2. Isolation of HAVEC, HSaVEC, HITAEC, and HMVEC

The study was conducted according to the Good Clinical Practice guidelines and the latest revision of the Declaration of Helsinki (2013), and the donor study protocol was ap-proved by the Local Ethical Committee of the Research Institute for Complex Issues of Cardiovascular Diseases (Kemerovo, Russia, protocol code: 7/2024, date of approval: 16 September 2024). Written informed consent was provided by all the study participants after receiving a full explanation of the study’s purposes.

HAVEC were isolated from aortic valve leaflets which were excised from patients with calcific aortic valve disease in a cardiac surgery unit #1 of the Research Institute for Complex Issues of Cardiovascular Diseases. Upon the excision, the tissues were immediately transferred into the cell culture laboratory in 15 mL tubes (601002, Wuxi NEST Biotechnology Co., Ltd., Wuxi, China) filled with EndoBoost medium (EB1, AppScience Products, Moscow, Russia) to preserve EC viability. Following the delivery, aortic valve leaflets were extensively washed in DPBS without Ca^2+^ and Mg^2+^ ions (pH = 7.4, 1.2.4.7, BioLot, St. Petersburg, Russia) and incubated for 10 min at 37 °C in 0.20% type IV collagenase solution (specific activity ≥ 160 U/mg, derived from *Clostridium histolyticum*, GC305015, Wuhan Servicebio Technology Co., Ltd., Wuhan, China). After vortexing the aortic valve for one minute to detach HAVEC, the flush was centrifuged at 300× *g* for five minutes, and HAVEC were seeded into T-25 cell culture flasks (07-9025, Biologix Plastic (Changzhou) Co., Ltd., Jinan, China) coated with 0.1% gelatin (1.4.6, BioLot, St. Petersburg, Russia) and filled with EndoBoost medium (EB1, AppScience Products, Moscow, Russia). The next day, we replaced EndoBoost medium (EB1, AppScience Products, Moscow, Russia) with EndoBoost Plus medium (EB2, AppScience Products, Moscow, Russia), and the cells were grown until reaching confluence.

HSaVEC, HITAEC, and HMVEC were isolated from saphenous vein (≈2 cm length) and internal thoracic artery segments (≈1 cm length) and fragments of subcutaneous fat (10 mL), which were excised from the patients who underwent coronary artery bypass graft surgery because of coronary artery disease, in cardiac surgery unit #1 of the Research Institute for Complex Issues of Cardiovascular Diseases. Upon the excision, the tissues were immediately transferred into the cell culture laboratory in 15 mL tubes (601002, Wuxi NEST Biotechnology Co., Ltd., Wuxi, China) filled with EndoBoost medium (EB1, AppScience Products, Moscow, Russia) to preserve EC viability. Then, the tissues were extensively washed in DPBS without Ca^2+^ and Mg^2+^ ions (pH = 7.4, 1.2.4.7, BioLot, St. Petersburg, Russia). Saphenous vein and internal thoracic artery segments were dissected longitudinally whilst adipose tissue fragments were cut into equal and small pieces, with the subsequent incubation in 15 mL tubes (601002, Wuxi NEST Biotechnology Co., Ltd., Wuxi, China) with 0.15% type IV collagenase solution (specific activity ≥ 160 U/mg, derived from *Clostridium histolyticum*, GC305015, Wuhan Servicebio Technology Co., Ltd., Wuhan, China) for 60 min at 37 °C. Following the incubation, the samples were slightly vortexed for 2 min for better shedding of ECs. Enzymatic digestion was halted by the addition of 5% fetal bovine serum (1.1.6.1, BioLot, St. Petersburg, Russia) dissolved in DPBS without Ca^2+^ and Mg^2+^ ions (pH = 7.4, 1.2.4.7, BioLot, St. Petersburg, Russia). The saphenous vein and internal thoracic artery segments were then discarded, whereas adipose tissue was removed by the cell strainers (70 µm pore diameter, 15-1070, Biologix Plastic (Changzhou) Co., Ltd., Jinan, China). The cells (HSaVEC, HITAEC, or HMVEC suspension) were centrifuged at 220× *g* for 5 min at room temperature and then seeded into T-25 cell culture flasks (07-9025, Biologix Plastic (Changzhou) Co., Ltd., Jinan, China) coated with fibronectin (10 µg/mL, 1.4.11, BioLot, St. Petersburg, Russia) and filled with EndoBoost medium (EB1, AppScience Products, Moscow, Russia). The next day, we replaced EndoBoost medium (EB1, AppScience Products, Moscow, Russia) with EndoBoost Plus medium (EB2, AppScience Products, Moscow, Russia), and the cells were grown until reaching confluence.

HAVEC, HSaVEC, HITAEC, and HMVEC were then purified using magnetic cell separation technique (EasySep magnet, STEMCELL Technologies, Vancouver, BC, Canada) and CD31^+^ microbeads (CD31 Dynabeads, 11155D, Thermo Fisher Scientific, Waltham, MA, USA) according to the manufacturer’s instructions. The purity of HAVEC, HSaVEC, HITAEC, and HMVEC was evaluated by flow cytometry, employing mouse phycoerythrin-cyanine 7-labeled anti-human CD146 (361008, BioLegend, San Diego, CA, USA), mouse fluorescein isothiocyanate-labeled anti-human CD31 (303104, BioLegend, San Diego, CA, USA), and mouse Alexa Fluor 700-labeled anti-human CD90 (328120, BioLegend, San Diego, CA, USA) monoclonal antibodies as well as respective isotype controls (400126, 400108, and 400144, BioLegend, San Diego, CA, USA). Cell cultures with ≥99.5% CD146^+^ CD90^−^ cells were treated as HAVEC, HSaVEC, HITAEC, and HMVEC and therefore were used for the subsequent experiments. The protocol for the isolation of these EC cultures has been described in detail in our previous papers [153,154]. Subculturing (i.e., passaging) was performed using EndoBoost medium (EB1, AppScience Products, Moscow, Russia), and EndoBoost Plus medium (EB2, AppScience Products, Moscow, Russia) was then used to grow cells until confluence in each passage. For the subculturing, we used T-25 or T-75 cell culture flasks (07-9025 or 07-8075, respectively, Biologix Plastic (Changzhou) Co., Ltd., Jinan, China) coated with 0.1% gelatin (1.4.6, BioLot, St. Petersburg, Russia).

### 4.3. Phase Contrast Microscopy and Sample Collection

HAVEC, HSaVEC, HITAEC, and HMVEC were seeded into T-25 flasks (*n* = 3 cultures per EC type, *n* = 12 cell cultures in total, 708003, Wuxi NEST Biotechnology Co., Ltd., Wuxi, China) and grown using EndoBoost Plus medium (EB2, AppScience Products, Moscow, Russia) until reaching confluence. Immediately before the experiments, we removed serum-supplemented EndoBoost Plus medium (EB2, AppScience Products, Moscow, Russia), washed the cells twice with warm (≈37 °C) DPBS, without Ca^2+^ and Mg^2+^ ions, (pH = 7.4, 1.2.4.7, BioLot, St. Petersburg, Russia) to remove the residual serum components, and then incubated the cells with serum-free EndoLife medium (EL1, AppScience Products, Moscow, Russia) containing 0.8 mmol/L BSA, PA-BSA, or OA-BSA for 24 h. The plasma concentrations of PA and OA in healthy individuals vary from 0.1 to 0.4 mmol/L [155], and exposure to their excessive levels (i.e., from 0.8 to 1.0 mmol/L) for from 4 to 24 h is typically employed in cell culture studies [156,157,158,159].

Following 24 h of exposure, we examined cells by phase contrast microscopy (AxioObserver.Z1, Carl Zeiss, Oberkochen, Germany), withdrew the cell culture medium into 15 mL tubes (601002, Wuxi NEST Biotechnology Co., Ltd., Wuxi, China), washed cells with ice-cold (4 °C) PBS (pH = 7.4, 2.1.1, BioLot, St. Petersburg, Russia), and lysed the cells in TRIzol reagent (15596018, Thermo Fisher Scientific, Waltham, MA, USA) to extract RNA according to the manufacturer’s protocols. The cell culture medium was centrifuged at 220× *g* (5804R, Eppendorf, Hamburg, Germany) to sediment detached cells and at 2000× *g* (MiniSpin Plus, Eppendorf, Hamburg, Germany) to sediment the cell debris. Then, the cell culture supernatant was transferred into the new low-protein binding tubes (Ac-ACT-017 L-B-S, Accumax Lab Devices Pvt. Ltd., Gujarat, India) and frozen at −80 °C (DW-86L486E, Haier Biomedical, Qingdao Haier Biomedical Co., Ltd., Qingdao, China) for the subsequent ELISA measurements.

### 4.4. WST Assay for Cell Proliferation and Viability

Confluent cultures of HAVEC, HSaVEC, HITAEC, and HMVEC were seeded into 96-well plates (701001, Wuxi NEST Biotechnology Co., Ltd., Wuxi, China) using serum-supplemented EndoBoost medium (EB1, AppScience Products, Moscow, Russia). The day after the seeding, we replaced the EndoBoost medium (EB1, AppScience Products, Moscow, Russia) with a serum-free EndoLife medium (EL1, AppScience Products, Moscow, Russia) containing 0.8 mmol/L BSA, PA-BSA, or OA-BSA for 24 h. Then, we replaced the medium with 100 µL of fresh serum-free EndoLife medium (EL1, AppScience Products, Moscow, Russia), added 10 µL of WST-8 reagent (G4103, Wuhan Servicebio Technology Co., Ltd., Wuhan, China) for 2 h, and conducted a colorimetric detection using a spectrophotometer (Multiskan Sky, Thermo Fisher Scientific, Waltham, MA, USA) at a 450 nm wavelength.

### 4.5. Gene Expression Analysis

Gene expression analysis in BSA-, PA-BSA-, and OA-BSA-treated HAVEC, HSaVEC, HITAEC, and HMVEC was performed by reverse transcription–polymerase chain reaction (RT-qPCR), as in our previous studies [56,160,161] (*n* = 3 patients per EC type, *n* = 12 patients in total). In each RT-qPCR measurement, we made three technical replicates for each patient in order to conduct an objective calculation of the arithmetic mean. Briefly, M-MuLV–RH First Strand cDNA Synthesis Kit (R01-250, Evrogen, Moscow, Russia) and reverse transcriptase M-MuLV–RH (R03-50, Evrogen, Moscow, Russia) was used for the reverse transcription, and RT-qPCR was carried out employing customized primers (500 nmol/L each, Evrogen, Moscow, Russia, Table 2), cDNA (20 ng), and BioMaster HS-qPCR Lo-ROX SYBR Master Mix (MHR031-2040, Biolabmix, Novosibirsk, Russia) according to the manufacturer’s protocol. The quantification of the mRNA levels (*VCAM1*, *ICAM1*, *SELE*, *SELP*, *MIF*, *PTX3*, *CSF2*, *CSF3*, *IL1A*, *IL6*, *CCL2*, *CCL5*, *CCL20*, *CSF2*, *CSF3*, *CXCL1*, *CXCL2*, *CXCL3*, *CXCL5*, *CXCL6*, *CXCL8*, *CXCL10*, *SNAI1*, *SNAI2*, *TWIST1*, *ZEB1*, *NOS3*, *VWF*, *SERPINE1*, *PLAU*, and *PLAT* genes) was performed by calculation of ΔCt and by using the 2^−ΔΔCt^ method. The relative transcript levels were expressed as a value relative to the average of *PECAM1* gene and to the mock-treated group (2^−ΔΔCt^).

### 4.6. Enzyme-Linked Immunosorbent Assay (ELISA)

Quantitative measurement of the IL-6, IL-8, and MCP-1/CCL2 levels in the cell culture supernatant was performed using the respective ELISA kits (A-8768, A-8762, and A-8782, Vector-Best, Novosibirsk, Russia) according to the manufacturer’s protocols (*n* = 2 technical aliquotes per cell culture sample; *n* = 12 samples per group). For each ELISA measurement, we loaded 100 µL of cell culture supernatant across all samples. The colorimetric detection of ELISA results was conducted using a spectrophotometer (Multiskan Sky, Thermo Fisher Scientific, Waltham, MA, USA) at a 450 nm wavelength.

### 4.7. Dot Blot Profiling

The semi-quantitative analyses of 109 cytokines and chemokines in the cell culture supernatant were performed using Proteome Profiler Human XL Cytokine Array Kits (ARY022B, R&D Systems, Minneapolis, MN, USA) according to the manufacturer’s instructions. In order to improve the sensitivity of dot blot profiling, all samples of cell culture supernatant were concentrated to equal volumes (tenfold, from 3330 μL to 333 μL per sample) utilizing a vacuum centrifugal concentrator (HyperVAC-LITE, Gyrozen, Gimpo, Republic of Korea). For each EC type, samples from three different patients were pooled into a single sample (i.e., 1 mL of concentrated cell culture supernatant per EC type) to obtain an average result. The chemiluminescent detection was conducted using the Odyssey XF imaging system (LI-COR Biosciences, Lincoln, NE, USA). The densitometric analysis of the chemiluminescent images was performed employing ImageJ software (version 1.54k, National Institutes of Health, Bethesda, MD, USA).

### 4.8. Measurement of Reactive Oxygen Species Generation

Human umbilical vein endothelial cells (HUVEC) obtained from three healthy donors (passage three) were cultured in gelatin-coated 96-well plates (µClear, Greiner Bio-One, Kremsmünster, Austria) for 5 days with a daily change in human microvascular endothelial cell medium (111-500, Cell Applications, San Diego, CA, USA) supplemented with 0.8 mmol/L BSA, PA-BSA, or OA-BSA. Prior to the ROS measurements, the medium was replaced with a fresh medium containing 10 µmol/L 2′,7′-dichlorodihydrofluorescein diacetate (DCF-DA, Sigma-Aldrich, Saint Louis, MO, USA) for 1 h and then with a Hank’s balanced salt solution without phenol red (P020p, PanEco, Moscow, Russia) containing 10 mmol/L of 4-(2-hydroxyethyl)-1-piperazineethanesulfonic acid (HEPES, F134E-100, PanEco, Moscow, Russia). The microplates were placed on an automated coordinate stage of the Axiovert 200 M fluorescent microscope equipped with an on-stage incubator and AxioCamHR camera (Carl Zeiss, Oberkochen, Germany). The DCF fluorescence was recorded from the same positions in wells every 20 min for a total period of 1 h (i.e., three times) using ×32 objective and fixed exposure settings. In parallel, phase contrast images of cells in the same fields of view were acquired to determine the number of cells producing the fluorescent signal. As PA-BSA reduced cell viability, ROS fluorescence was normalized by the number of survived cells. The data were expressed as an increase in DCF fluorescence over 1 h per cell.

### 4.9. Rescue Experiments

Confluent cultures of HAVEC, HSaVEC, HITAEC, and HMVEC were seeded into 96-well plates (701001, Wuxi NEST Biotechnology Co., Ltd., Wuxi, China) using serum-supplemented EndoBoost medium (EB1, AppScience Products, Moscow, Russia). The day after the seeding, we replaced the EndoBoost medium (EB1, AppScience Products, Moscow, Russia) with a serum-free EndoLife medium (EL1, AppScience Products, Moscow, Russia) and treated ECs with 0.8 mmol/L of BSA, PA-BSA, or OA-BSA for 24 h, with or without the following: (1) antioxidant enzymes superoxide dismutase (SOD, 250 U/mL, S5395, Sigma-Aldrich, Saint Louis, MO, USA) and catalase (CAT, 500 U/mL, SRE0041, Sigma-Aldrich, Saint Louis, MO, USA); (2) antioxidant cell culture supplement L-ascorbic acid 2-phosphate (L-AA2P, 50 µg/mL, A2521, TCI Chemicals, Tokyo, Japan); and (3) antioxidant cell culture additive sodium selenite (NaSe, 7 µg/mL, F080, PanEco, Moscow, Russia). The abovementioned antioxidants were added to ECs 3 h before their exposure to BSA, PA-BSA, or OA-BSA, to ensure their internalization through the plasma membrane. After 24 h of incubation, we replaced the medium with 100 µL of fresh serum-free EndoLife medium (EL1, AppScience Products, Moscow, Russia), added 10 µL WST-8 reagent (G4103, Wuhan Servicebio Technology Co., Ltd., Wuhan, China) for 2 h, and conducted a colorimetric detection using a spectrophotometer (Multiskan Sky, Thermo Fisher Scientific, Waltham, MA, USA) at a 450 nm wavelength.

### 4.10. Statistical Analysis

Statistical analysis was performed using GraphPad Prism 8 (GraphPad Software, San Diego, CA, USA). For the WST-8 assay, data are presented as median, 25th and 75th percentiles, and range, and the groups were compared by the Mann–Whitney U-test (in the case of pairwise comparisons) or by the Kruskal–Wallis test with a subsequent Dunn’s multiple comparisons test to perform a respective adjustment. For RT-qPCR analysis, data are presented as an arithmetic mean and standard deviation, and groups were compared by a ratio paired *t*-test (i.e., ratios of paired values but not absolute differences between paired values were compared) because of the considerable between-sample variability in absolute values and selected metric (fold change). For the ELISA measurements, data are presented as median, 25th and 75th percentiles, and range, and groups were compared by the Wilcoxon matched-pairs signed-rank test because of the considerable between-sample variability in absolute values. *p*-Values ≤ 0.05 were regarded as statistically significant.

## 5. Conclusions

PA-BSA is detrimental for ECs and provokes an evident pro-inflammatory response in all EC types, primarily HAVEC and HSaVEC. Although OA-BSA exerts moderate cytotoxic effects on HAVEC and HMVEC, it does not induce generalized EC activation. Across the four studied EC types (HAVEC, HSaVEC, HITAEC, and HMVEC), HITAEC was the most resistant to PA-BSA exposure and was not affected by OA-BSA. Collectively, these results suggest significant heterogeneity in the response to PA-BSA in the ECs from different vascular beds.

## Figures and Tables

**Figure 1 ijms-26-12148-f001:**
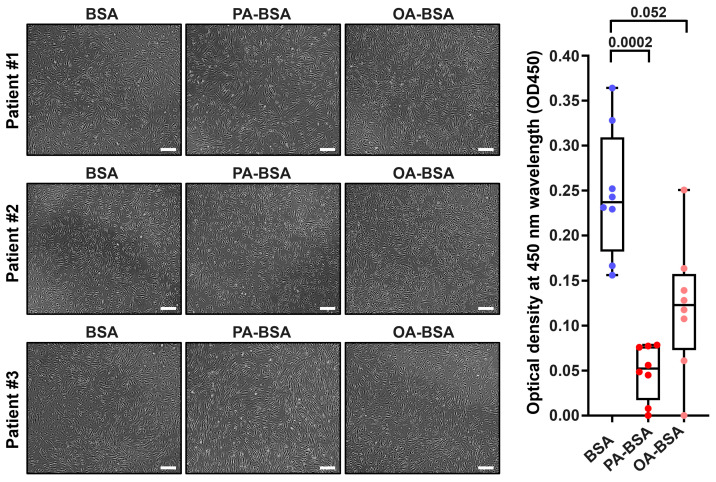
Cytotoxic effects of conjugates of palmitic acid (PA) with a fatty acid-free BSA (PA-BSA, 0.8 mmol/L) or conjugates of oleic acid (OA) with a fatty acid-free BSA (OA-BSA, 0.8 mmol/L) on primary human aortic valve endothelial cells (HAVEC) after the 24 h incubation. Fatty acid-free bovine serum albumin (BSA) was used as a control group. (**Left**): Phase contrast microscopy, representative images. Magnification: ×200. Scale bar: 100 µm. (**Right**): Microplate colorimetric analysis of proliferation and viability (WST-8 assay). Each dot on the plots represents an arithmetic mean of eight replicates (i.e., 8 wells of a 96-well plate) for each patient (*n* = 12 patients, *n* = 3 per each of the indicated EC types). Box-and-whisker plot. Whiskers indicate the range, box bounds indicate the 25th–75th percentiles, and center lines indicate the median. *p*-Values are provided above the boxes, via a Kruskal–Wallis test with a subsequent Dunn’s multiple comparisons test.

**Figure 2 ijms-26-12148-f002:**
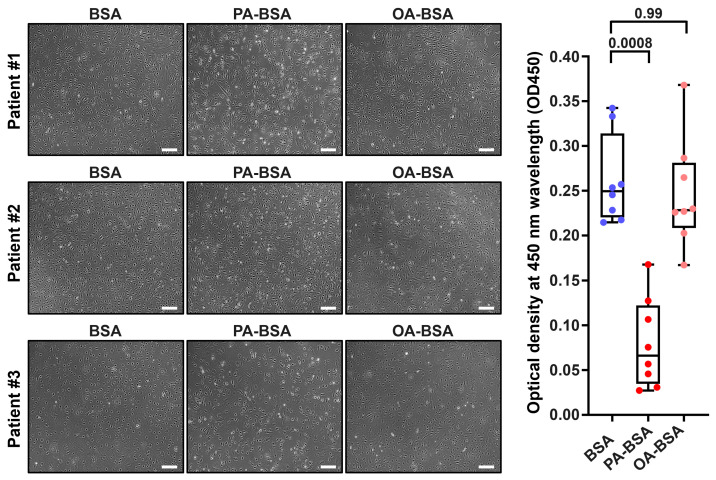
Cytotoxic effects of conjugates of palmitic acid (PA) with a fatty acid-free BSA (PA-BSA, 0.8 mmol/L) or conjugates of oleic acid (OA) with a fatty acid-free BSA (OA-BSA, 0.8 mmol/L) on primary human saphenous vein endothelial cells (HSaVEC) after the 24 h incubation. Fatty acid-free bovine serum albumin (BSA) was used as a control group. (**Left**): Phase contrast microscopy; representative images. Magnification: ×200. Scale bar: 100 µm. (**Right**): Microplate colorimetric analysis of proliferation and viability (WST-8 assay). Each dot on the plots represents an arithmetic mean of eight replicates (i.e., 8 wells of a 96-well plate) for each patient (*n* = 12 patients, *n* = 3 per each of the indicated EC types). Box-and-whisker plot. Whiskers indicate the range, box bounds indicate the 25th–75th percentiles, and center lines indicate the median. *p*-Values are provided above the boxes, via a Kruskal–Wallis test with a subsequent Dunn’s multiple comparisons test.

**Figure 3 ijms-26-12148-f003:**
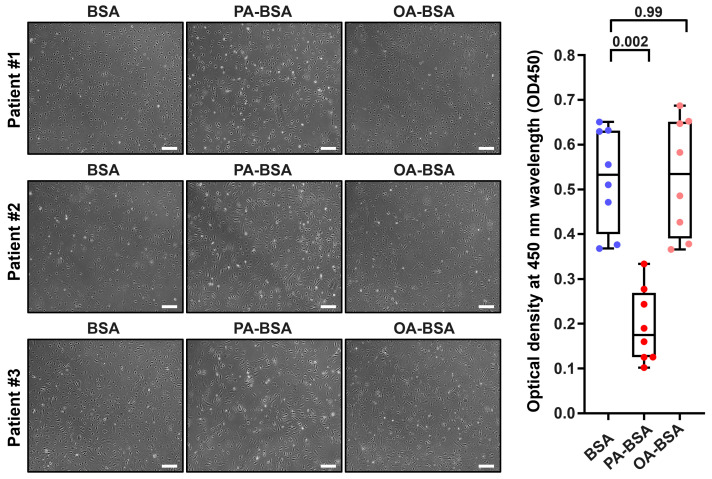
Cytotoxic effects of conjugates of palmitic acid (PA) with a fatty acid-free BSA (PA-BSA, 0.8 mmol/L) or conjugates of oleic acid (OA) with a fatty acid-free BSA (OA-BSA, 0.8 mmol/L) on primary human internal thoracic artery endothelial cells (HITAEC) after the 24 h incubation. Fatty acid-free bovine serum albumin (BSA) was used as a control group. (**Left**): Phase contrast microscopy; representative images. Magnification: ×200. Scale bar: 100 µm. (**Right**): Microplate colorimetric analysis of proliferation and viability (WST-8 assay). Each dot on the plots represents an arithmetic mean of eight replicates (i.e., 8 wells of a 96-well plate) for each patient (*n* = 12 patients, *n* = 3 per each of the indicated EC types). Box-and-whisker plot. Whiskers indicate the range, box bounds indicate the 25th–75th percentiles, and center lines indicate the median. *p*-Values are provided above boxes, via Kruskal–Wallis test with subsequent Dunn’s multiple comparisons test.

**Figure 4 ijms-26-12148-f004:**
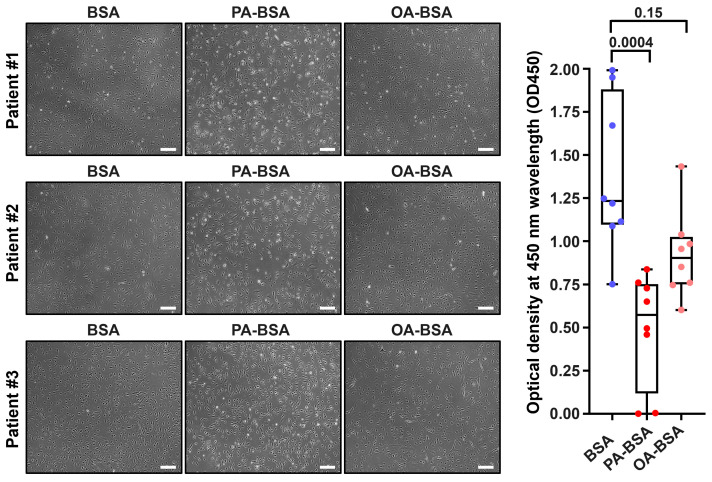
Cytotoxic effects of conjugates of palmitic acid (PA) with a fatty acid-free BSA (PA-BSA, 0.8 mmol/L) or conjugates of oleic acid (OA) with a fatty acid-free BSA (OA-BSA, 0.8 mmol/L) on primary human adipose tissue-derived microvascular endothelial cells (HMVEC) after the 24 h incubation. Fatty acid-free bovine serum albumin (BSA) was used as a control group. (**Left**): Phase contrast microscopy; representative images. Magnification: ×200. Scale bar: 100 µm. (**Right**): Microplate colorimetric analysis of proliferation and viability (WST-8 assay). Each dot on the plots represents an arithmetic mean of eight replicates (i.e., 8 wells of a 96-well plate) for each patient (*n* = 12 patients, *n* = 3 per each of the indicated EC types). Box-and-whisker plot. Whiskers indicate the range, box bounds indicate the 25th–75th percentiles, and center lines indicate the median. *p*-Values are provided above the boxes, via the Kruskal–Wallis test with subsequent Dunn’s multiple comparisons test.

**Figure 5 ijms-26-12148-f005:**
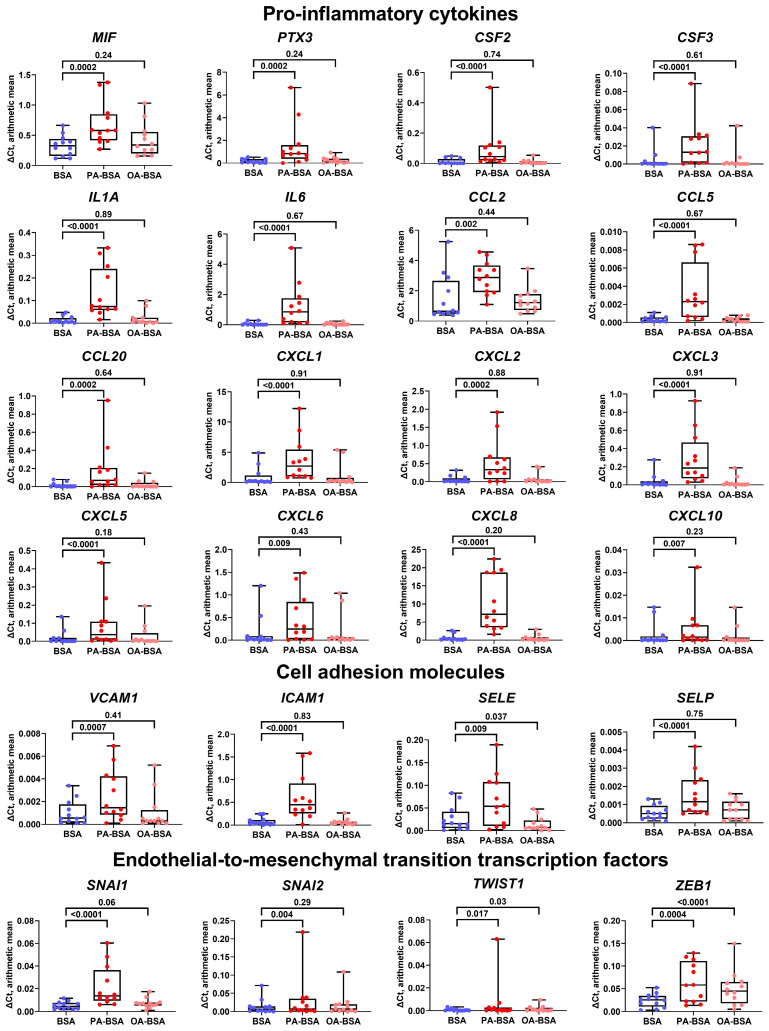
Relative levels of gene expression (arithmetic mean) in genes encoding pro-inflammatory cytokines (*MIF*, *PTX3*, *CSF2*, *CSF3*, *IL1A*, *IL6*, *CCL2*, *CCL5*, *CCL20*, *CXCL1*, *CXCL2*, *CXCL3*, *CXCL5*, *CXCL6*, *CXCL8*, and *CXCL10*), pro-inflammatory cell adhesion molecules (*VCAM1*, *ICAM1*, *SELE*, and *SELP*), and endothelial-to-mesenchymal transition transcription factors (*SNAI1*, *SNAI2*, *TWIST1*, and *ZEB1*) in primary human aortic valve endothelial cells (HAVEC), primary human saphenous vein endothelial cells (HSaVEC), primary human internal thoracic artery endothelial cells (HITAEC), and primary human adipose tissue-derived microvascular endothelial cells (HMVEC) incubated with control fatty acid-free bovine serum albumin (BSA), conjugates of palmitic acid (PA) with a fatty acid-free BSA (PA-BSA, 0.8 mmol/L), or conjugates of oleic acid (OA) with a fatty acid-free BSA (OA-BSA, 0.8 mmol/L) for 24 h. Box-and-whisker plot. Whiskers indicate the range; *p*-values are provided above the boxes, via a ratio paired *t*-test (i.e., ratios of paired values but not absolute differences between paired values were compared).

**Figure 6 ijms-26-12148-f006:**
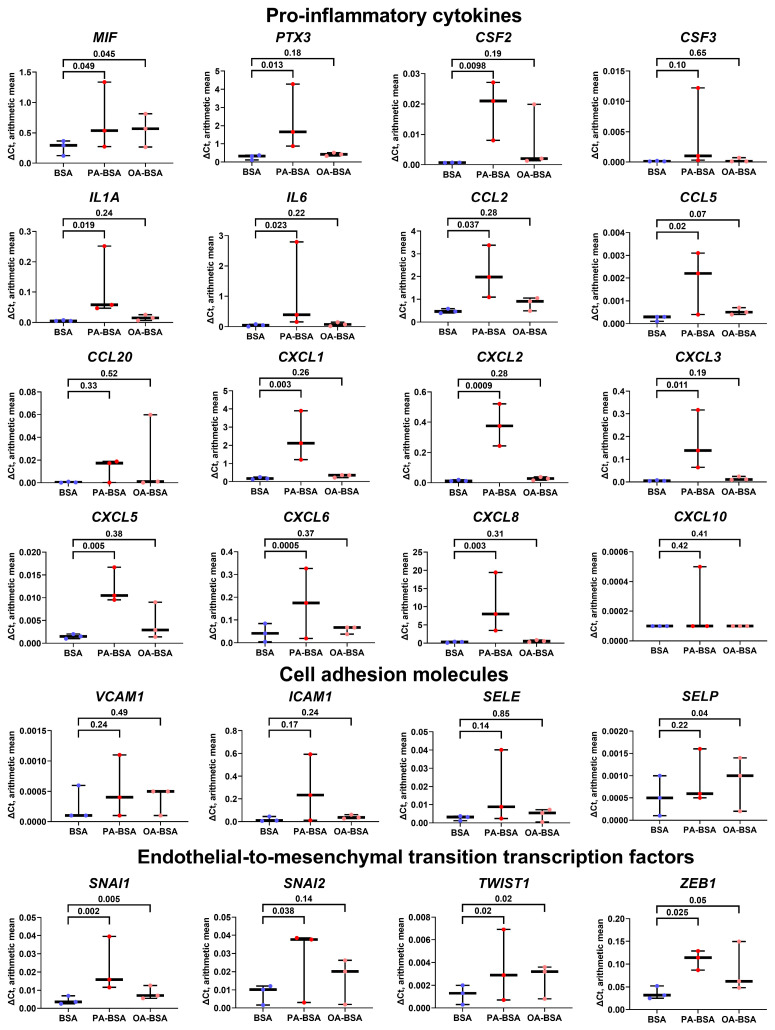
Relative levels of gene expression (arithmetic mean) in genes encoding pro-inflammatory cytokines (*MIF*, *PTX3*, *CSF2*, *CSF3*, *IL1A*, *IL6*, *CCL2*, *CCL5*, *CCL20*, *CXCL1*, *CXCL2*, *CXCL3*, *CXCL5*, *CXCL6*, *CXCL8*, and *CXCL10*), pro-inflammatory cell adhesion molecules (*VCAM1*, *ICAM1*, *SELE*, and *SELP*), and endothelial-to-mesenchymal transition transcription factors (*SNAI1*, *SNAI2*, *TWIST1*, and *ZEB1*) in primary human aortic valve endothelial cells (HAVEC) incubated with control fatty acid-free bovine serum albumin (BSA), conjugates of palmitic acid (PA) with a fatty acid-free BSA (PA-BSA, 0.8 mmol/L), or conjugates of oleic acid (OA) with a fatty acid-free BSA (OA-BSA, 0.8 mmol/L) for 24 h. Box-and-whisker plot. Whiskers indicate the range; *p*-values are provided above the boxes, via a ratio paired *t*-test (i.e., ratios of paired values but not absolute differences between paired values were compared).

**Figure 7 ijms-26-12148-f007:**
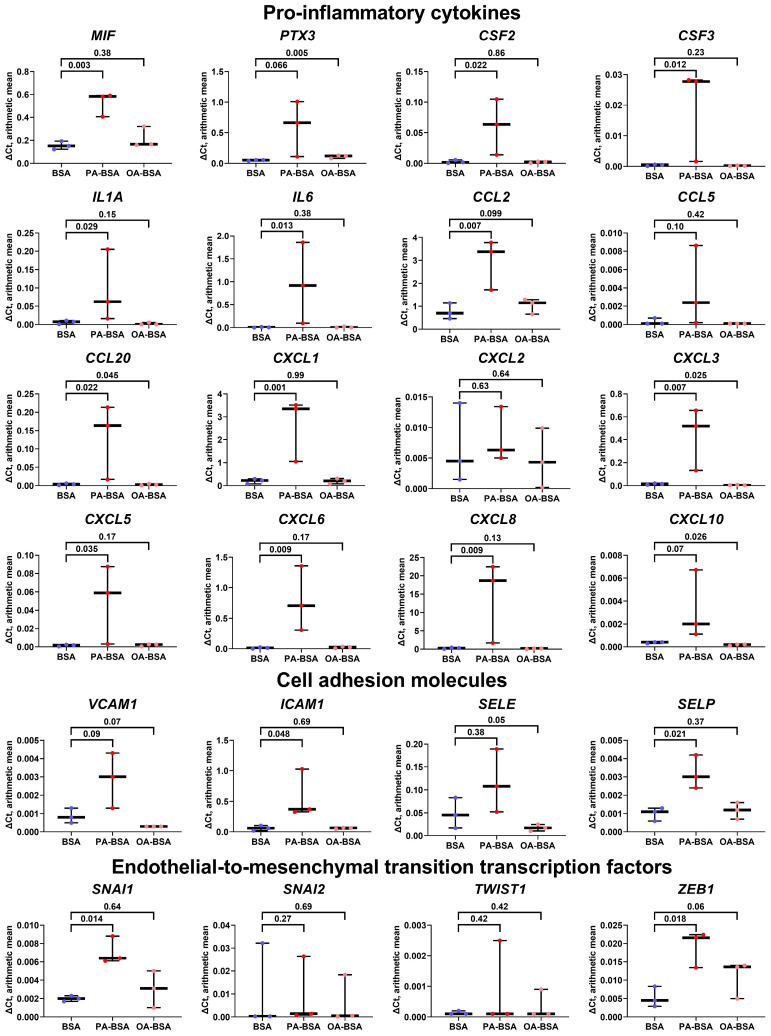
Relative levels of gene expression (arithmetic mean) in genes encoding pro-inflammatory cytokines (*MIF*, *PTX3*, *CSF2*, *CSF3*, *IL1A*, *IL6*, *CCL2*, *CCL5*, *CCL20*, *CXCL1*, *CXCL2*, *CXCL3*, *CXCL5*, *CXCL6*, *CXCL8*, and *CXCL10*), pro-inflammatory cell adhesion molecules (*VCAM1*, *ICAM1*, *SELE*, and *SELP*), and endothelial-to-mesenchymal transition transcription factors (*SNAI1*, *SNAI2*, *TWIST1*, and *ZEB1*) in primary human saphenous vein endothelial cells (HSaVEC) incubated with control fatty acid-free bovine serum albumin (BSA), conjugates of palmitic acid (PA) with a fatty acid-free BSA (PA-BSA, 0.8 mmol/L), or conjugates of oleic acid (OA) with a fatty acid-free BSA (OA-BSA, 0.8 mmol/L) for 24 h. Box-and-whisker plot. Whiskers indicate the range; *p*-values are provided above the boxes, via a ratio paired *t*-test (i.e., ratios of paired values but not absolute differences between paired values were compared).

**Figure 8 ijms-26-12148-f008:**
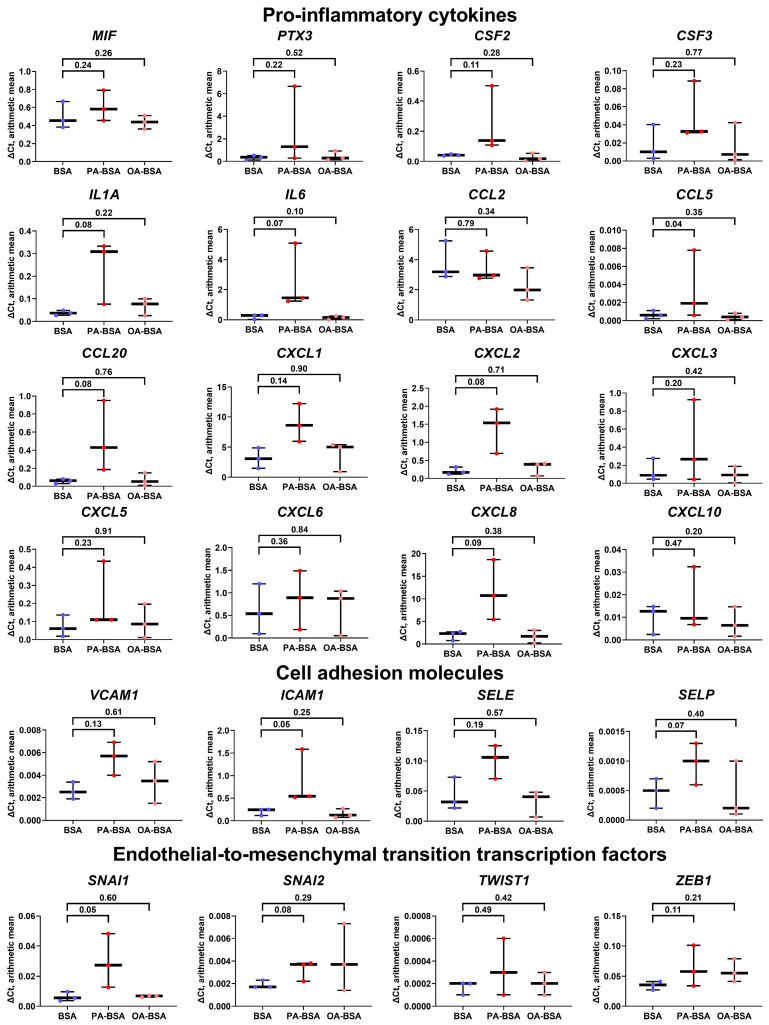
Relative levels of gene expression (arithmetic mean) in genes encoding pro-inflammatory cytokines (*MIF*, *PTX3*, *CSF2*, *CSF3*, *IL1A*, *IL6*, *CCL2*, *CCL5*, *CCL20*, *CXCL1*, *CXCL2*, *CXCL3*, *CXCL5*, *CXCL6*, *CXCL8*, and *CXCL10*), pro-inflammatory cell adhesion molecules (*VCAM1*, *ICAM1*, *SELE*, and *SELP*), and endothelial-to-mesenchymal transition transcription factors (*SNAI1*, *SNAI2*, *TWIST1*, and *ZEB1*) in primary human internal thoracic artery endothelial cells (HITAEC) incubated with control fatty acid-free bovine serum albumin (BSA), conjugates of palmitic acid (PA) with a fatty acid-free BSA (PA-BSA, 0.8 mmol/L), or conjugates of oleic acid (OA) with a fatty acid-free BSA (OA-BSA, 0.8 mmol/L) for 24 h. Box-and-whisker plot. Whiskers indicate the range; *p*-values are provided above boxes, via a ratio paired *t*-test (i.e., ratios of paired values but not absolute differences between paired values were compared).

**Figure 9 ijms-26-12148-f009:**
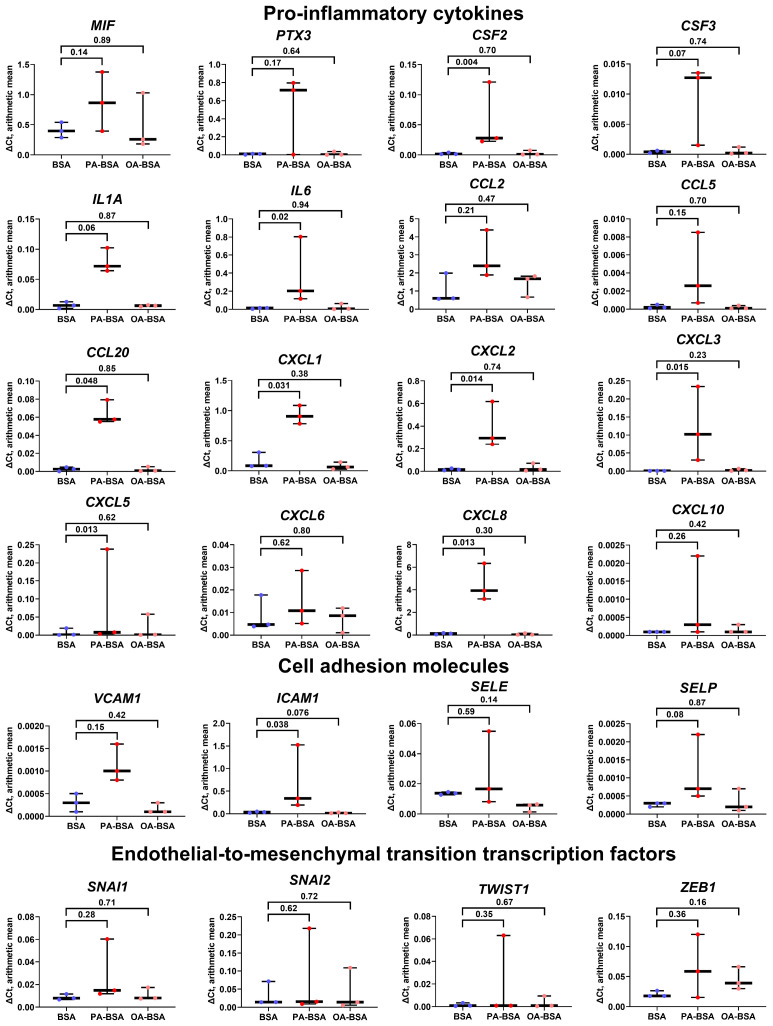
Relative levels of gene expression (arithmetic mean) in genes encoding pro-inflammatory cytokines (*MIF*, *PTX3*, *CSF2*, *CSF3*, *IL1A*, *IL6*, *CCL2*, *CCL5*, *CCL20*, *CXCL1*, *CXCL2*, *CXCL3*, *CXCL5*, *CXCL6*, *CXCL8*, and *CXCL10*), pro-inflammatory cell adhesion molecules (*VCAM1*, *ICAM1*, *SELE*, and *SELP*), and endothelial-to-mesenchymal transition transcription factors (*SNAI1*, *SNAI2*, and *TWIST1*, *ZEB1*) in primary human adipose tissue-derived microvascular endothelial cells (HMVEC) incubated with control fatty acid-free bovine serum albumin (BSA), conjugates of palmitic acid (PA) with a fatty acid-free BSA (PA-BSA, 0.8 mmol/L), or conjugates of oleic acid (OA) with a fatty acid-free BSA (OA-BSA, 0.8 mmol/L) for 24 h. Box-and-whisker plot. Whiskers indicate the range; *p*-values are provided above boxes, via a ratio paired *t*-test (i.e., ratios of paired values but not absolute differences between paired values were compared).

**Figure 10 ijms-26-12148-f010:**
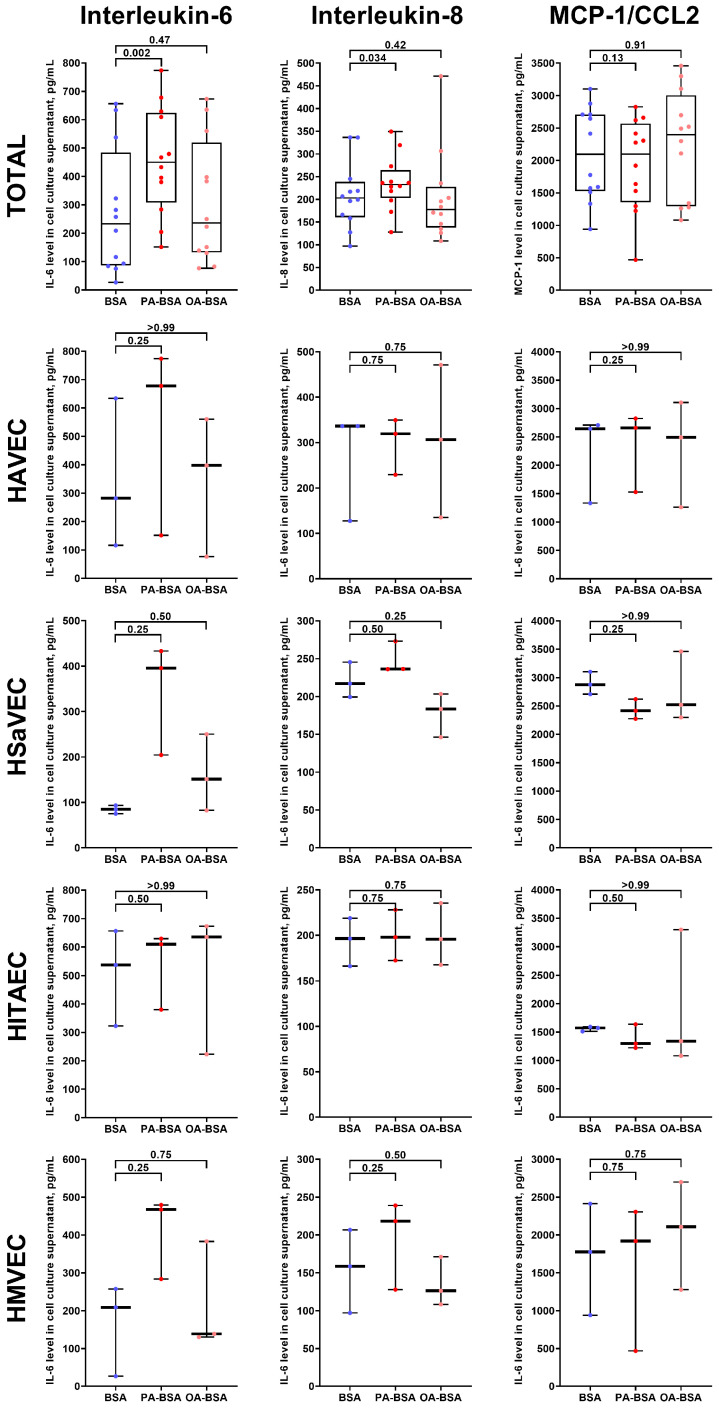
Enzyme-linked immunosorbent assay measurements of interleukin-6, interleukin-8, and MCP-1/CCL2 in pre-centrifuged (2000× *g*) serum-free cell culture supernatant from primary human aortic valve endothelial cells (HAVEC), human saphenous vein endothelial cells (HSaVEC), human internal thoracic artery endothelial cells (HITAEC), and human microvascular endothelial cells (HMVEC) incubated with control fatty acid-free bovine serum albumin (BSA), conjugates of palmitic acid (PA) with a fatty acid-free BSA (PA-BSA, 0.8 mmol/L), or conjugates of oleic acid (OA) with a fatty acid-free BSA (OA-BSA, 0.8 mmol/L) for 24 h. Each dot on the plots represents an arithmetic mean of two replicates (i.e., 8 wells of a 96-well plate) for each patient (*n* = 12 patients, *n* = 3 per each of the indicated EC types). Box-and-whisker plot. Whiskers indicate the range, box bounds indicate the 25th–75th percentiles, and center lines indicate the median. *p*-Values are provided above the boxes, via Wilcoxon matched-pairs signed-rank test.

**Figure 11 ijms-26-12148-f011:**
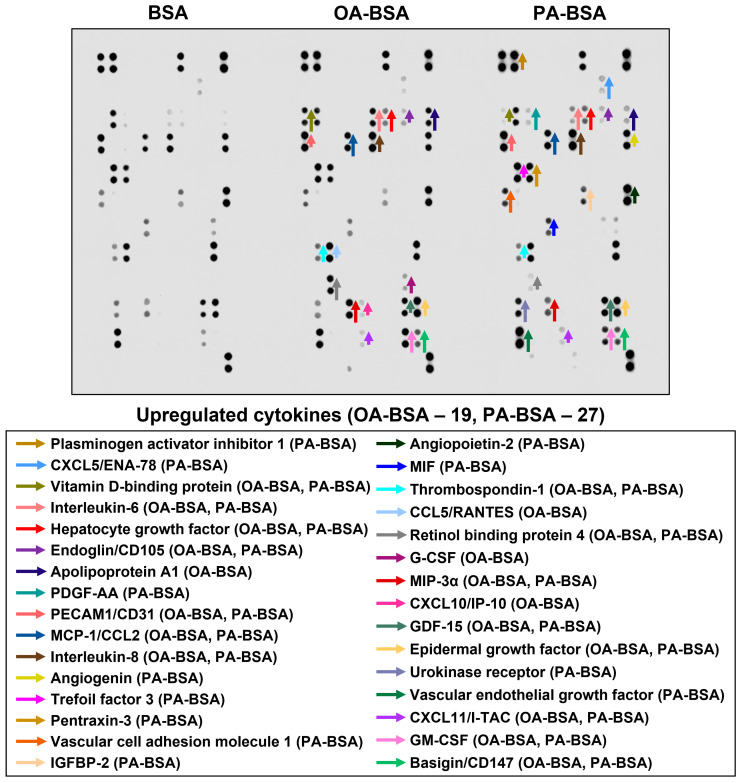
Semi-quantitative dot blot assessment of pro-inflammatory cytokines in pre-centrifuged (2000× *g*), tenfold-enriched serum-free cell culture supernatant withdrawn from primary human saphenous vein endothelial cells (HSaVEC) incubated with control fatty acid-free bovine serum albumin (BSA), conjugates of oleic acid (OA) with a fatty acid-free BSA (OA-BSA, 0.8 mmol/L), or conjugates of palmitic acid (PA) with a fatty acid-free BSA (PA-BSA, 0.8 mmol/L) for 24 h. The intensity of colored black dots directly reflects the expression level of the corresponding protein. The list of cytokines overrepresented in the cell culture supernatant (with the corresponding color-coded arrows) is shown below the figure. Arrow length represents the fold change upon densitometric analysis of chemiluminescent images, performed in the ImageJ software (version 1.54k). Short, medium, and long arrows represent fold changes of 1.20–1.34, 1.35–1.49, and ≥1.50, respectively, relative to densitometric values in the BSA group. In cases where protein was not expressed in the BSA group but was detected in OA-BSA or PA-BSA groups, densitometric values ≤ 4999, from 5000 to 9999, and ≥10,000 arbitrary units were denoted with short, medium, and long arrows, respectively.

**Figure 12 ijms-26-12148-f012:**
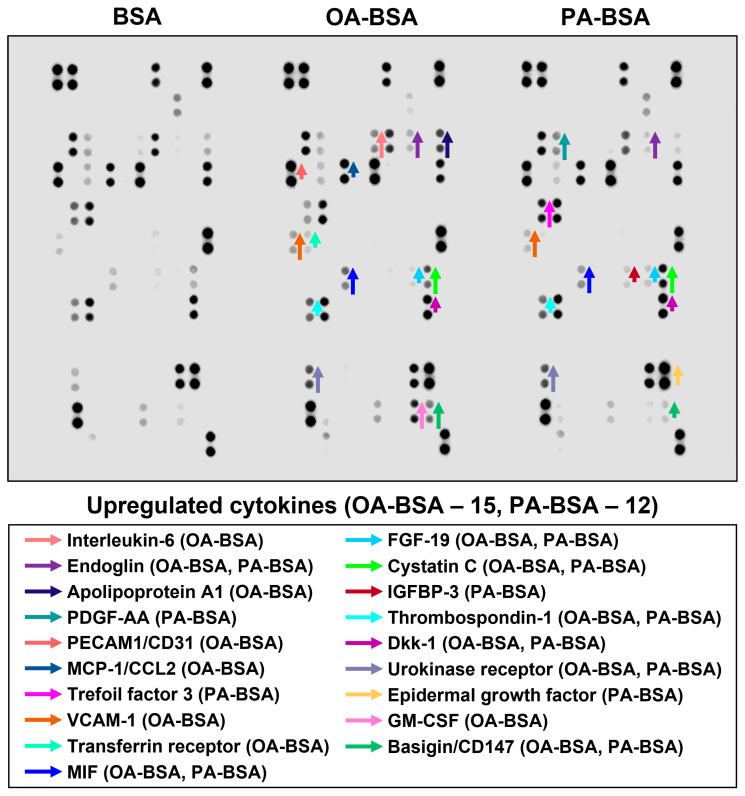
Semi-quantitative dot blot assessment of pro-inflammatory cytokines in pre-centrifuged (2000× *g*), tenfold-enriched serum-free cell culture supernatant withdrawn from primary human microvascular endothelial cells (HMVEC) incubated with control fatty acid-free bovine serum albumin (BSA), conjugates of oleic acid (OA) with a fatty acid-free BSA (OA-BSA, 0.8 mmol/L), or conjugates of palmitic acid (PA) with a fatty acid-free BSA (PA-BSA, 0.8 mmol/L) for 24 h. The intensity of colored black dots directly reflects the expression level of the corresponding protein. The list of cytokines overrepresented in the cell culture supernatant (with the corresponding color-coded arrows) is shown below the figure. Arrow length represents the fold change upon densitometric analysis of chemiluminescent images, performed in the ImageJ software (version 1.54k). Short, medium, and long arrows represent fold changes of 1.20–1.34, 1.35–1.49, and ≥1.50, respectively, relative to densitometric values in the BSA group. In cases where protein was not expressed in the BSA group but was detected in OA-BSA or PA-BSA groups, densitometric values ≤ 4999, from 5000 to 9999, and ≥10,000 arbitrary units were denoted with short, medium, and long arrows, respectively.

**Figure 13 ijms-26-12148-f013:**
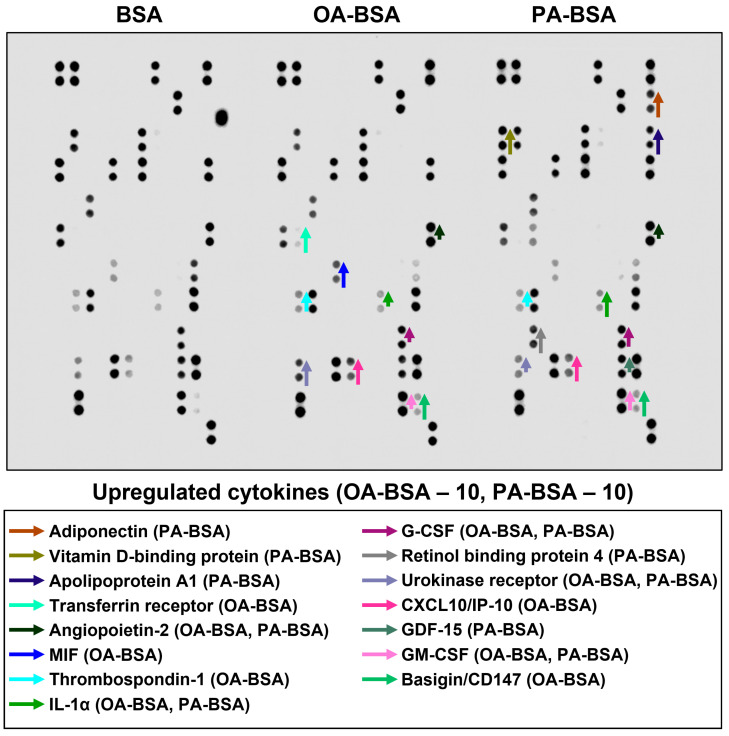
Semi-quantitative dot blot assessment of pro-inflammatory cytokines in pre-centrifuged (2000× *g*), tenfold-enriched serum-free cell culture supernatant withdrawn from primary human internal thoracic artery endothelial cells (HITAEC) incubated with control fatty acid-free bovine serum albumin (BSA), conjugates of oleic acid (OA) with a fatty acid-free BSA (OA-BSA, 0.8 mmol/L), or conjugates of palmitic acid (PA) with a fatty acid-free BSA (PA-BSA, 0.8 mmol/L) for 24 h. The intensity of colored black dots directly reflects the expression level of the corresponding protein. The list of cytokines overrepresented in the cell culture supernatant (with the corresponding color-coded arrows) is shown below the figure. Arrow length represents the fold change upon densitometric analysis of chemiluminescent images, performed in the ImageJ software (version 1.54k). Short, medium, and long arrows represent fold changes of 1.20–1.34, 1.35–1.49, and ≥1.50, respectively, relative to densitometric values in the BSA group. In cases where protein was not expressed in the BSA group but was detected in OA-BSA or PA-BSA groups, densitometric values ≤ 4999, from 5000 to 9999, and ≥10,000 arbitrary units were denoted with short, medium, and long arrows, respectively.

**Figure 14 ijms-26-12148-f014:**
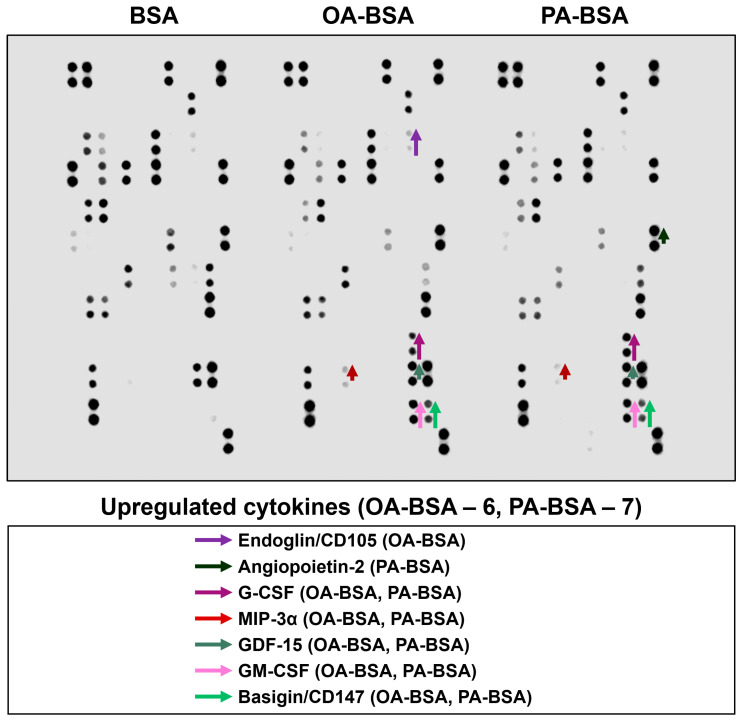
Semi-quantitative dot blot assessment of pro-inflammatory cytokines in pre-centrifuged (2000× *g*), tenfold-enriched serum-free cell culture supernatant withdrawn from primary human aortic valve endothelial cells (HAVEC) incubated with control fatty acid-free bovine serum albumin (BSA), conjugates of oleic acid (OA) with a fatty acid-free BSA (OA-BSA, 0.8 mmol/L), or conjugates of palmitic acid (PA) with a fatty acid-free BSA (PA-BSA, 0.8 mmol/L) for 24 h. The intensity of colored black dots directly reflects the expression level of the corresponding protein. The list of cytokines overrepresented in the cell culture supernatant (with the corresponding color-coded arrows) is shown below the figure. Arrow length represents the fold change upon densitometric analysis of chemiluminescent images, performed in the ImageJ software (version 1.54k). Short, medium, and long arrows represent fold changes of 1.20–1.34, 1.35–1.49, and ≥1.50, respectively, relative to densitometric values in the BSA group. In cases where protein was not expressed in the BSA group but was detected in OA-BSA or PA-BSA groups, densitometric values ≤ 4999, from 5000 to 9999, and ≥10,000 arbitrary units were denoted with short, medium, and long arrows, respectively.

**Figure 15 ijms-26-12148-f015:**
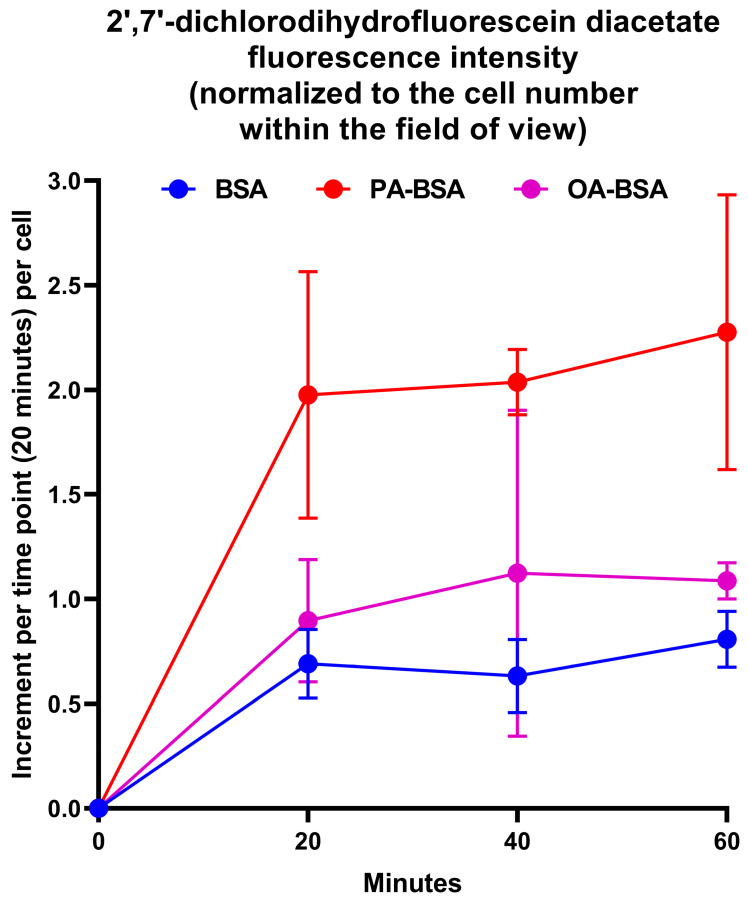
Measurement of reactive oxygen species (ROS)’s generation in human umbilical vein endothelial cells (HUVEC) incubated with control fatty acid-free bovine serum albumin (BSA), conjugates of palmitic acid (PA) with a fatty acid-free BSA (PA-BSA, 0.8 mmol/L), or conjugates of oleic acid (OA) with a fatty acid-free BSA (OA-BSA, 0.8 mmol/L) for 120 h. ROS generation in the ECs was detected by 2′,7′-dichlorodihydrofluorescein diacetate (DCF-DA) staining after PA-BSA or OA-BSA challenge. DCF fluorescence was recorded from the same positions in wells every 20 min for a total period of 1 h (i.e., three times) with fixed exposure settings, and was normalized to the cell number within the field of view. Each dot on the plots represents an arithmetic mean of three replicates (i.e., three wells of a 96-well plate), while whiskers indicate the standard deviation.

**Figure 16 ijms-26-12148-f016:**
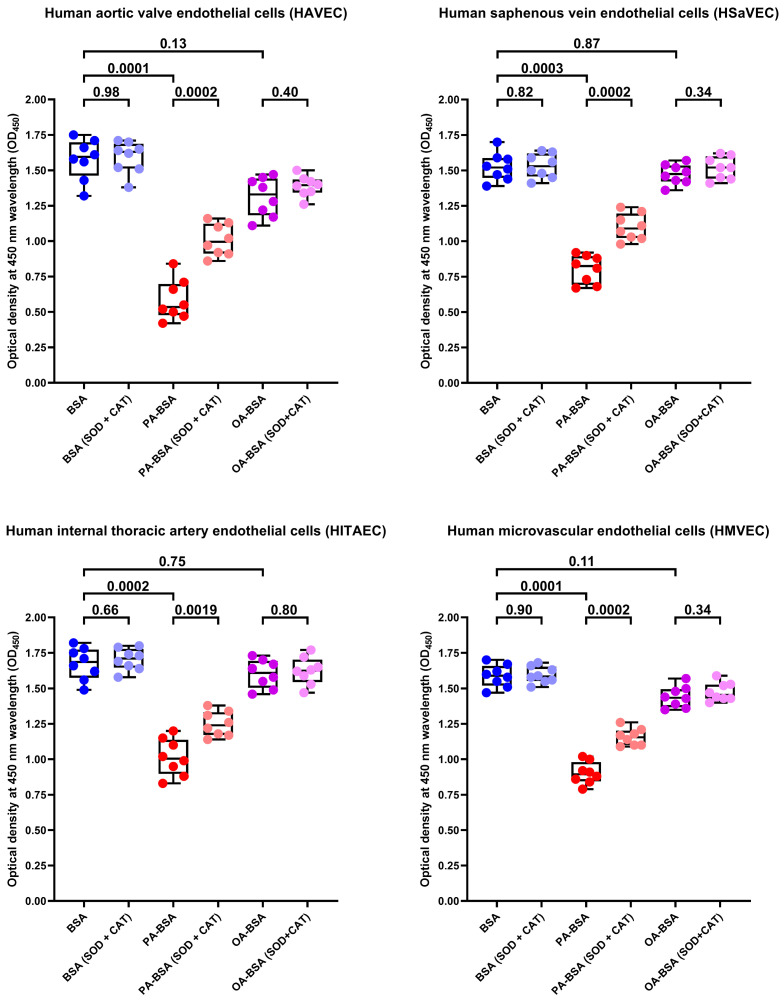
Superoxide dismutase (SOD, 250 U/mL) and catalase (CAT, 500 U/mL) rescue endothelial cells (ECs) from palmitic acid (PA)-induced death. Microplate colorimetric analysis of proliferation and viability (WST-8 assay) of primary human aortic valve endothelial cells (HAVEC), human saphenous vein endothelial cells (HSaVEC), human internal thoracic artery endothelial cells (HITAEC), and human microvascular endothelial cells (HMVEC) incubated with control fatty acid-free bovine serum albumin (BSA), conjugates of palmitic acid (PA) with a fatty acid-free BSA (PA-BSA, 0.8 mmol/L), or conjugates of oleic acid (OA) with a fatty acid-free BSA (OA-BSA, 0.8 mmol/L) for 24 h. Each dot on the plots represents an arithmetic mean of eight replicates (i.e., 8 wells of a 96-well plate). Box-and-whisker plot. Whiskers indicate the range, box bounds indicate the 25th–75th percentiles, and center lines indicate the median. *p*-Values are provided above boxes. The Mann–Whitney U-test was used to analyze the pairwise comparisons with and without superoxide dismutase and catalase treatment, and the Kruskal–Wallis test with subsequent Dunn’s multiple comparisons test were used for analyzing the comparisons of BSA with PA-BSA and OA-BSA.

**Figure 17 ijms-26-12148-f017:**
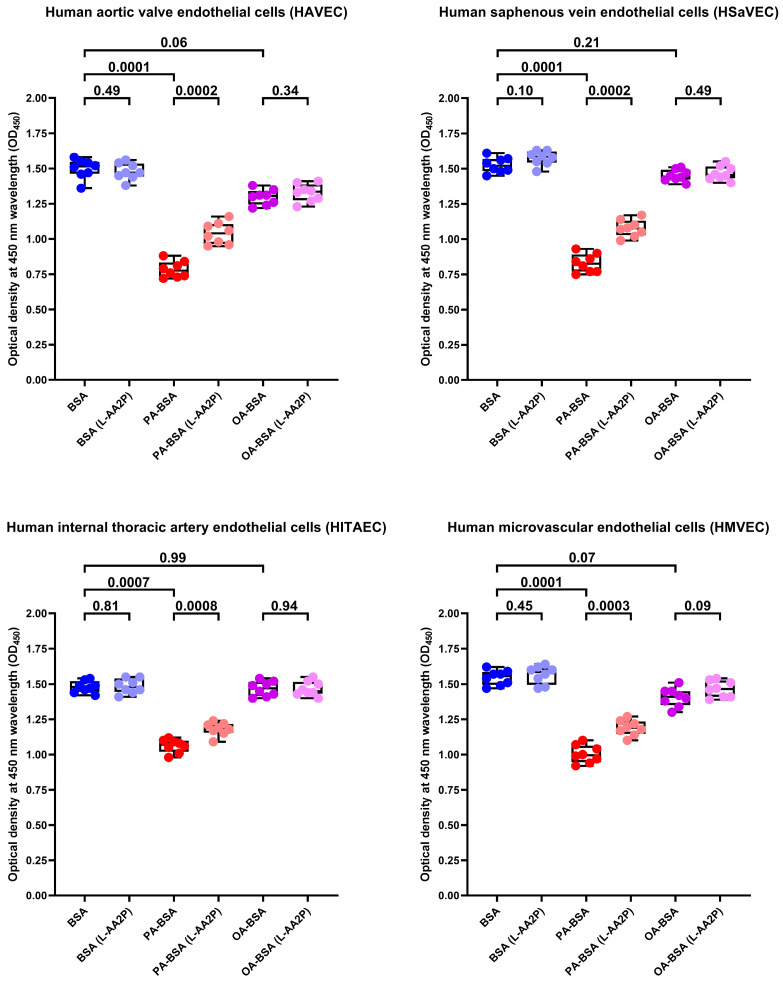
L-ascorbic acid 2-phosphate (L-AA2P, 50 µg/mL) rescues endothelial cells (ECs) from palmitic acid (PA)-induced death. Microplate colorimetric analysis of proliferation and viability (WST-8 assay) of primary human aortic valve endothelial cells (HAVEC), human saphenous vein endothelial cells (HSaVEC), human internal thoracic artery endothelial cells (HITAEC), and human microvascular endothelial cells (HMVEC) incubated with control fatty acid-free bovine serum albumin (BSA), conjugates of palmitic acid (PA) with a fatty acid-free BSA (PA-BSA, 0.8 mmol/L), or conjugates of oleic acid (OA) with a fatty acid-free BSA (OA-BSA, 0.8 mmol/L) for 24 h. Each dot on the plots represents an arithmetic mean of eight replicates (i.e., 8 wells of a 96-well plate). Box-and-whisker plot. Whiskers indicate the range, box bounds indicate the 25th–75th percentiles, and center lines indicate the median. *p*-Values are provided above the boxes. The Mann–Whitney U-test was used to analyze the pairwise comparisons with and without L-ascorbic acid 2-phosphate treatment, and the Kruskal–Wallis test with subsequent Dunn’s multiple comparisons test were used for analyzing the comparisons of BSA with PA-BSA and OA-BSA.

**Figure 18 ijms-26-12148-f018:**
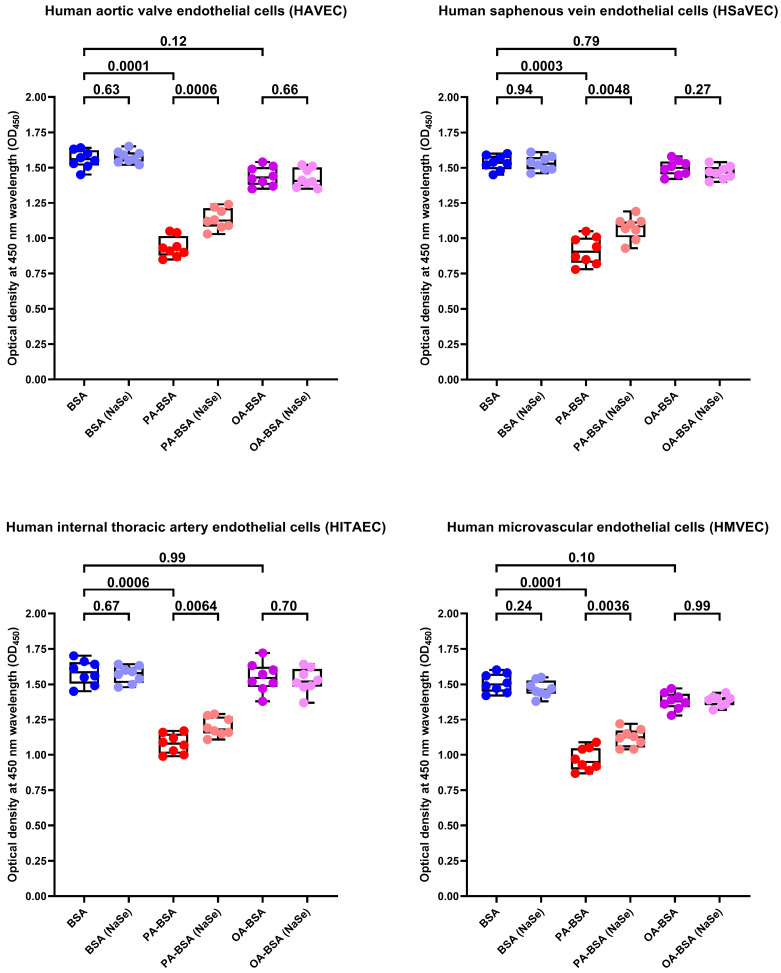
Sodium selenite (NaSe, 7 µg/mL) rescues endothelial cells (ECs) from palmitic acid (PA)-induced death. Microplate colorimetric analysis of proliferation and viability (WST-8 assay) of primary human aortic valve endothelial cells (HAVEC), human saphenous vein endothelial cells (HSaVEC), human internal thoracic artery endothelial cells (HITAEC), and human microvascular endothelial cells (HMVEC) incubated with control fatty acid-free bovine serum albumin (BSA), conjugates of palmitic acid (PA) with a fatty acid-free BSA (PA-BSA, 0.8 mmol/L), or conjugates of oleic acid (OA) with a fatty acid-free BSA (OA-BSA, 0.8 mmol/L) for 24 h. Each dot on the plots represents an arithmetic mean of eight replicates (i.e., 8 wells of a 96-well plate). Box-and-whisker plot. Whiskers indicate the range, box bounds indicate the 25th–75th percentiles, and center lines indicate the median. *p*-Values are provided above boxes. The Mann–Whitney U-test was used to analyze the pairwise comparisons with and without sodium selenite treatment, and the Kruskal–Wallis test with subsequent Dunn’s multiple comparisons test were used for analyzing the comparisons of BSA with PA-BSA and OA-BSA.

**Table 1 ijms-26-12148-t001:** Densitometric semi-quantitative analysis of cytokine levels (dot blot profiling with chemiluminescent detection) was performed in the pre-centrifuged (2000× *g*), tenfold-enriched, serum-free culture medium derived from primary human saphenous vein endothelial cells (HSaVEC), human microvascular endothelial cells (HMVEC), human internal thoracic artery endothelial cells (HITAEC), and human aortic valve endothelial cells (HAVEC) incubated with control fatty acid-free bovine serum albumin (BSA), conjugates of oleic acid (OA) with a fatty acid-free BSA (OA-BSA, 0.8 mmol/L), or conjugates of palmitic acid (PA) with a fatty acid-free BSA (PA-BSA, 0.8 mmol/L) for 24 h. Densitometric analysis was performed using the ImageJ software (version 1.54k). Fold changes of 1.20–1.34, 1.35–1.49, and ≥1.50 (relative to densitometric values in the BSA group, respectively) correspond to short, medium, and long arrows in Figure 11, Figure 12, Figure 13 and Figure 14. In cases where protein was not expressed in the control group but was detected in OA-BSA or PA-BSA groups, densitometric values ≤ 4999, from 5000 to 9999, and ≥10,000 arbitrary units, respectively, correspond to short, medium, and long arrows in Figure 11, Figure 12, Figure 13 and Figure 14.

Location on the Membrane	Analyte	BSA	OA-BSA	Fold Change	PA-BSA	Fold Change
Human saphenous vein endothelial cells (HSaVEC)						
A5,6	Apolipoprotein A1	708	22,494	31.78	6244	8.82
A7,8	Angiogenin	24,976	25,195	1.01	31,622	1.27
A11,12	Angiopoietin-2	33,403	32,467	0.97	46,048	1.38
B13,14	Cystatin C	8585			4977	0.58
B15,16	Dickkopf-related protein 1	33,459	32,276	0.96	36,140	1.08
B19,20	Epidermal growth factor	27,825	38,830	1.40	38,199	1.37
B21,22	Basigin/CD147	4654	12,164	2.61	13,549	2.91
C3,4	Chemokine (C–X–C motif) ligand 5/Epithelial neutrophil-activating protein 78	15,216	12,803	0.84	24,412	1.60
C5,6	Endoglin/CD105		4322	4322	3155	3155
C17,18	Granulocyte colony-stimulating factor		5626	5626		
C19,20	Growth/differentiation factor 15	17,690	23,369	1.32	29,984	1.69
C21,22	Granulocyte–macrophage colony-stimulating factor		29,238	29,238	17,854	17,854
D1,2	Chemokine (C–X–C motif) ligand 1/Growth-regulated oncogene α	25,626	28,634	1.12	26,717	1.04
D5,6	Hepatocyte growth factor	1114	9741	8.74	12,292	11.03
D11,12	Insulin-like growth factor-binding protein 2	7915	5710	0.72	13,468	1.70
E5,6	Interleukin-6	2059	24,607	11.95	6628	3.22
E7,8	Interleukin-8	24,979	36,636	1.47	43,670	1.75
F19,20	Chemokine (C–X–C motif) ligand 10/Interferon gamma-induced protein 10		2423	2423		
F21,22	Chemokine (C–X–C motif) ligand 11/Interferon-inducible t-cell alpha chemoattractant		2187	2187	3072	3072
G7,8	Monocyte chemoattractant protein 1	16,406	25,348	1.55	27,986	1.71
G13,14	Macrophage migration inhibitory factor	17,649	15,426	0.87	26,164	1.48
G19,20	Macrophage inflammatory protein-3α	10,383	77,802	7.49	15,892	1.53
H5,6	Platelet-derived growth factor, composed of two A subunits	820			3107	3.79
H9,10	Pentraxin-3	20,600	24,006	1.17	32,095	1.56
H15,16	Chemokine (C–C motif) ligand 5/Regulated upon activation, normal T cell expressed, and secreted	26,970	35,553	1.32	29,749	1.10
H17,18	Retinol binding protein 4		33,324	33,324	2429	2429
I1,2	Serpin E1/Plasminogen activator inhibitor-1	32,950	37,290	1.13	45,188	1.37
I5,6	Suppression of tumorigenicity 2	23,005	20,214	0.88	24,080	1.05
I9,10	Trefoil factor 3	30,586	30,816	1.01	37,906	1.24
I15,16	Thrombospondin-1	7327	10,813	1.48	9432	1.29
I19,20	Urokinase receptor	8830	9494	1.08	23,610	2.67
I21,22	Vascular endothelial growth factor	28,368	30,060	1.06	50,268	1.77
J5,6	Vitamin D-binding protein		20,544	20,544	1183	1183
J7,8	Platelet endothelial cell adhesion molecule-1/CD31	29,205	35,281	1.21	39,773	1.36
J11,12	Vascular cell adhesion protein 1	8473	9463	1.12	16,304	1.92
Human microvascular endothelial cells (HMVEC)						
A5,6	Apolipoprotein A1	4941	19,026	3.85	2121	0.43
A7,8	Angiogenin	28,435	23,933	0.84	30,259	1.06
A11,12	Angiopoietin-2	49,197	49,009	1.00	37,446	0.76
B13,14	Cystatin C	8068	15,685	1.94	32,759	4.06
B15,16	Dickkopf-related protein 1	30,025	35,356	1.18	39,990	1.33
B19,20	Epidermal growth factor	40,745	47,634	1.17	55,697	1.37
B21,22	Basigin/CD147		10,316	10,316	2549	2549
C3,4	Chemokine C–X–C motif) ligand 5/Epithelial neutrophil-activating protein 78	30,330	5560	0.18	24,625	0.81
C5,6	Endoglin/CD105	460	4939	10.75	2239	4.87
C13,14	Fibroblast growth factor-19		1270	1270	3577	3577
C19,20	Growth/differentiation factor 15	31,906	27,132	0.85	34,489	1.08
C21,22	Granulocyte–macrophage colony-stimulating factor	1521	18,819	12.37	877	0.58
D1,2	Chemokine (C–X–C) ligand 1/Growth-regulated oncogene α	24,702	28,720	1.16	24,389	0.99
D5,6	Hepatocyte growth factor	23,493	23,784	1.01	18,792	0.80
D13,14	Insulin-like growth factor-binding protein 3				3150	3150
E5,6	Interleukin-6	1046	10,621	10.15		
E7,8	Interleukin-8	38,331	45,555	1.19	45,083	1.18
E21,22	Interleukin-17a	5145	5567	1.08	3329	0.65
G7,8	Monocyte chemoattractant protein 1	24,687	31,297	1.27	28,427	1.15
G13,14	Macrophage migration inhibitory factor	5499	14,614	2.66	10,545	1.92
H5,6	Platelet-derived growth factor, composed of two A subunits	6650	5718	0.86	10,787	1.62
H7,8	Platelet-derived growth factor, composed of A and B or two B subunits	4318	3769	0.87	2617	0.61
H9,10	Pentraxin-3	30,987	33,459	1.08	35,450	1.14
H15,16	Chemokine (C–C) ligand 5/Regulated upon activation, normal T cell expressed, and secreted	28,187	31,018	1.10	32,734	1.16
I1,2	Serpin E1	41,141	45,628	1.11	40,514	0.98
I5,6	Suppression of tumorigenicity 2	29,998	32,742	1.09	30,392	1.01
I9,10	Trefoil factor 3	16,638	9608	0.58	26,550	1.60
I11,12	Transferrin receptor		2460	2460		
I15,16	Thrombospondin-1	11,834	15,414	1.30	15,137	1.28
I19,20	Urokinase receptor	5330	15,487	2.91	14,675	2.75
I21,22	Vascular endothelial growth factor	42,687	44,673	1.05	46,020	1.08
J7,8	Platelet endothelial cell adhesion molecule-1/CD31	36,342	46,007	1.27	38,125	1.05
J11,12	Vascular cell adhesion protein 1	2359	6458	2.74	3688	1.56
Human internal thoracic artery endothelial cells (HITAEC)						
A3,4	Adiponectin				22,669	22,669
A5,6	Apolipoprotein A1				23,125	23,125
A7,8	Angiogenin	33,169	28,838	0.87	35,768	1.08
A11,12	Angiopoietin-2	31,310	40,163	1.28	38,626	1.23
B13,14	Cystatin C	28,793	6973	0.24	15,130	0.53
B15,16	Dickkopf-related protein 1	38,310	36,670	0.96	31,388	0.82
B19,20	Epidermal growth factor	41,406	40,552	0.98	41,469	1.00
B21,22	Basigin/CD147	1588	5800	3.65	6975	4.39
C3,4	Chemokine (C–X–C motif) ligand 5/Epithelial neutrophil-activating protein 78	35,527	36,747	1.03	34,561	0.97
C17,18	Granulocyte colony-stimulating factor	24,672	30,176	1.22	33,310	1.35
C19,20	Growth/differentiation factor 15	21,036	21,413	1.02	26,338	1.25
C21,22	Granulocyte–macrophage colony-stimulating factor	29,570	39,667	1.34	40,004	1.35
D1,2	Chemokine (C–X–C motif) ligand 1/Growth-regulated oncogene α	28,629	30,861	1.08	31,020	1.08
D15,16	Interleukin-1α	5231	6784	1.30	7867	1.50
E5,6	Interleukin-6	32,060	28,641	0.89	31,076	0.97
E7,8	Interleukin-8	34,304	33,969	0.99	40,088	1.17
F19,20	Chemokine (C–X–C motif) ligand 10/Interferon gamma-induced protein 10	7751	16,545	2.13	15,477	2.00
G7,8	Monocyte chemoattractant protein 1	26,790	25,809	0.96	24,625	0.92
G13,14	Macrophage migration inhibitory factor	11,970	24,067	2.01	10,117	0.85
G19,20	Macrophage inflammatory protein-3α	33,694	37,162	1.10	35,738	1.06
H9,10	Pentraxin-3	29,532	21,859	0.74	28,106	0.95
H15,16	Chemokine (C–C motif) ligand 5/Regulated upon activation, normal T cell expressed, and secreted	26,336	29,885	1.13	29,683	1.13
H17,18	Retinol binding protein 4				24,834	24,834
I1,2	Serpin E1/Plasminogen activator inhibitor-1	41,010	37,687	0.92	40,644	0.99
I5,6	Suppression of tumorigenicity 2	27,915	16,560	0.59	25,667	0.92
I11,12	Transferrin receptor		15,034	15,034		
I15,16	Thrombospondin-1	6427	9353	1.46	7860	1.22
I19,20	Urokinase receptor	15,097	28,527	1.89	17,901	1.19
I21,22	Vascular endothelial growth factor	41,811	45,097	1.08	44,888	1.07
J5,6	Vitamin D-binding protein				36,030	36,030
J7,8	Platelet endothelial cell adhesion molecule-1/CD31	35,387	34,402	0.97	30,813	0.87
J11,12	Vascular cell adhesion protein 1	25,835	16,857	0.65	18,949	0.73
Human aortic valve endothelial cells (HAVEC)						
A7,8	Angiogenin	37,221	31,160	0.84	33,416	0.90
A11,12	Angiopoietin-2	36,796	43,772	1.19	44,475	1.21
B13,14	Cystatin C	29,617	7449	0.25	18,692	0.63
B15,16	Dickkopf-related protein 1	42,330	40,481	0.96	36,586	0.86
B19,20	Epidermal growth factor	42,948	45,228	1.05	48,909	1.14
B21,22	Basigin/CD147		15,873	15,873	12,962	12,962
C3,4	Chemokine (C–X–C motif) ligand 5/Epithelial neutrophil-activating protein 78	29,239	25,628	0.88	28,887	0.99
C5,6	Endoglin/CD105	10,275	17,577	1.71	6085	0.59
C17,18	Granulocyte colony-stimulating factor		20,111	20,111	33,590	33,590
C19,20	Growth/differentiation factor 15	25,009	30,013	1.20	30,783	1.23
C21,22	Granulocyte–macrophage colony-stimulating factor		28,037	28,037	25,440	25,440
D1,2	Chemokine (C–X–C motif) ligand 1/Growth-regulated oncogene α	32,560	34,263	1.05	31,538	0.97
D11,12	Insulin-like growth factor-binding protein 2	16,578	6377	0.38	6723	0.41
D13,14	Insulin-like growth factor-binding protein 3	7615				
E5,6	Interleukin-6	36,735	30,185	0.82	32,317	0.88
E7,8	Interleukin-8	41,077	36,177	0.88	36,858	0.90
G7,8	Monocyte chemoattractant protein 1	30,133	25,629	0.85	26,803	0.89
G13,14	Macrophage migration inhibitory factor	24,785	25,859	1.04	10,824	0.44
G19,20	Macrophage inflammatory protein 3α		3302	3302	1343	1343
H5,6	Platelet-derived growth factor, composed of two A subunits	5854	632	0.11	1376	0.24
H7,8	Platelet-derived growth factor, composed of A and B or two B subunits	16,812	7818	0.47	7121	0.42
H9,10	Pentraxin-3	34,548	28,644	0.83	34,171	0.99
H15,16	Chemokine (C–C motif) ligand 5/Regulated upon activation, normal T cell expressed, and secreted	12,561	11,846	0.94	12,323	0.98
I1,2	Serpin E1	44,492	36,145	0.81	45,135	1.01
I5,6	Suppression of tumorigenicity 2	18,996	8947	0.47	12,055	0.63
I9,10	Trefoil factor 3	23,557	6388	0.27	11,619	0.49
I15,16	Thrombospondin-1	17,631	19,138	1.09	13,058	0.74
I17,18	Tumor necrosis factor α					
I19,20	Urokinase receptor	19,916	20,838	1.05	22,339	1.12
I21,22	Vascular endothelial growth factor	44,242	51,034	1.15	48,465	1.10
J7,8	Platelet endothelial cell adhesion molecule-1/CD31	43,180	39,363	0.91	38,797	0.90
J11,12	Vascular cell adhesion protein 1	2570	1018	0.40	956	0.37

**Table 2 ijms-26-12148-t002:** Primer sequences for reverse transcription–quantitative polymerase chain reaction.

Gene	Forward Primer	Reverse Primer
Pro-inflammatory cytokines		
*MIF*	5′-GGTGTCCGAGAAGTCAGGCA-3′	5′-GGGGCACGTTGGTGTTTACG-3′
*PTX3*	5′-GAACTTTGCGTCTCTCCAGCAA-3′	5′-AGAGCTTGTCCCATTCCGAGT-3′
*CSF2*	5′-AGCCTCACCAAGCTCAAGGG-3′	5′-GGGGATGACAAGCAGAAAGTCC-3′
*CSF3*	5′-TCCAGGAGAAGCTGGTGAGTGA-3′	5′-GAGCCCCTGGTAGAGGAAAAGG-3′
*IL1A*	5′-TGCCCAAGATGAAGACCAACC-3′	5′-AGTGCCGTGAGTTTCCCAGAA-3′
*IL6*	5′-GGCACTGGCAGAAAACAACC-3′	5′-GCAAGTCTCCTCATTGAATCC-3′
*CCL2*	5′-TTCTGTGCCTGCTGCTCATAG-3′	5′-AGGTGACTGGGGCATTGATTG-3′
*CCL5*	5′-AGTGGCAAGTGCTCCAACCC-3′	5′-TCAGCCGGGAGTCATACAGGA-3′
*CCL20*	5′-TGAAGGCTGTGACATCAATGCT-3′	5′-CCATTCCAGAAAAGCCACAGTT-3′
*CXCL1*	5′-GCTTGCCTCAATCCTGCATCC-3′	5′-ACAATCCAGGTGGCCTCTGC-3′
*CXCL2*	5′-TTGTCTCAACCCCGCATCG-3′	5′-TGCTCAAACACATTAGGCGCAA-3′
*CXCL3*	5′-TGCGCCCAAACCGAAGTCAT-3′	5′-TCAGCTCTGGTAAGGGCAGGGA-3′
*CXCL5*	5′-ATCTGCAAGTGTTCGCCATAGG-3′	5′-TCCATGCGTGCTCATTTCTCTT-3′
*CXCL6*	5′-CTGCGTTGCACTTGTTTACGC-3′	5′-GCTTCCGGGTCCAGACAAACT-3′
*CXCL8*	5′-CAGAGACAGCAGAGCACAC-3′	5′-AGTTCTTTAGCACTCCTTGGC-3′
*CXCL10*	5′-AGGAACCTCCAGTCTCAGCAC-3′	5′-GGACAAAATTGGCTTGCAGGA-3′
Pro-inflammatory cell adhesion molecules		
*VCAM1*	5′-CGTCTTGGTCAGCCCTTCCT-3′	5′-ACATTCATATACTCCCGCATCCTTC-3′
*ICAM1*	5′-TTGGGCATAGAGACCCCGTT-3′	5′-GCACATTGCTCAGTTCATACACC-3′
*SELE*	5′-GCACAGCCTTGTCCAACC-3′	5′-ACCTCACCAAACCCTTCG-3′
*SELP*	5′-ATGGGTGGGAACCAAAAAGG-3′	5′-GGCTGACGGACTCTTGATGTAT-3′
Endothelial-to-mesenchymal transition transcription factors		
*SNAI1*	5′-CAGACCCACTCAGATGTCAAGAA-3′	5′-GGGCAGGTATGGAGAGGAAGA-3′
*SNAI2*	5′-ACTCCGAAGCCAAATGACAA-3′	5′-CTCTCTCTGTGGGTGTGTGT-3′
*TWIST1*	5′-GTCCGCAGTCTTACGAGGAG-3′	5′-GCTTGAGGGTCTGAATCTTGCT-3′
*ZEB1*	5′-GATGATGAATGCGAGTCAGATGC-3′	5′-ACAGCAGTGTCTTGTTGTTGT-3′
Housekeeping genes		
*PECAM1*	5′-AAGGAACAGGAGGGAGAGTATTA-3′	5′-GTATTTTGCTTCTGGGGACACT-3′

## Data Availability

The raw data supporting the conclusions of this article will be made available by the authors on request.

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
