# Peer review of "Palmitic but Not Oleic Acid Induces Pro-Inflammatory Dysfunction of Human Endothelial Cells from Different Vascular Beds In Vitro"

_ijms, 2025, doi:10.3390/ijms262412148_

Round 1

Reviewer 1 Report (Previous Reviewer 1)

Comments and Suggestions for Authors

The Authors have satisfactorily addressed all my previous comments. They provided a clear explanation of the cytotoxic effects observed under bright-field microscopy, and presenting the patients’ bright-field images with the cytotoxicity results separately for the different districts has made the data easier to interpret. They also refined the interpretation of the OA-BSA effects on HMVECs. The use of violin plots for the qRealTime PCR results has greatly improved their readability, and the inclusion of additional dot blot, oxidative stress, and recovery assays has further strengthened the study. The discussion has also been substantially improved. Based on these revisions, the manuscript is suitable for publication in its current form.

Author Response

We sincerely thank the reviewer for the high evaluation of our study.

Reviewer 2 Report (New Reviewer)

Comments and Suggestions for Authors In general,  the study is rigorously executed that provides useful insights into the differential effects of fatty acids on endothelial cell function. A particular strength is the parallel investigation across four distinct primary human endothelial cell types, allowing for a direct and informative comparison of vascular bed-specific responses. The negative effects of palmitic acid contrast with the more benign profile of oleic acid well. There are some minor points that the authors should address. First, why choosing the dosage used in the test? Second, how do the authors suggest situation in vivo? Third, how are PA and OA absorbed and metabolized, there should be some brief introduction, this is linked to the 2nd point.                   

Author Response

We sincerely thank the reviewer for the constructive criticism and valuable suggestions, which significantly helped us to improve the manuscript. In the attached file, we provide a point-by-point response to the reviewer’s suggestions. Please see the attachment.

This manuscript is a resubmission of an earlier submission. The following is a list of the peer review reports and author responses from that submission.

Round 1

Reviewer 1 Report

Comments and Suggestions for Authors

Shishkova et al. present an in vitro study on primary endothelial cell cultures derived from different districts, exposed to palmitic and oleic acid treatments in order to evaluate their effects on various parameters. The topic is consistent with the scope of the journal, and the introduction is adequate in guiding the reader into the subject. Nevertheless, several observations need to be addressed.

line 115: From Figure 1, it is really difficult for the reader to clearly appreciate the differences between the treatments. In addition, the Authors should better specify what is meant by the cytotoxic effects mentioned, as these are currently not well defined.

Figure 2, HMVEC panel: as observed for HAVEC, survival also appears to be reduced in OA-BSA. Therefore, the Authors should revise the corresponding paragraph accordingly.

Table 1: it would be advisable to replace or complement it with box-and-whisker plots, organized by topic (e.g., cytokines, adhesion molecules, endothelial-to-mesenchymal transitions, etc.), as these are easier to consult and provide clearer and more direct information.

For the qPCR analysis, how many patients were included and how many replicates were performed? I could not find this information.

The section from lines 231 to 268 is inappropriate and should be substantially revised. The Authors are expected to provide a critical discussion of the implications of their findings in the context of the existing literature, rather than presenting a descriptive list of molecules antagonizing the effects of palmitic acid. The approach adopted from line 270 onwards is more appropriate and should serve as the model for the discussion.

lines 330-346: Do the authors know the conjugation efficiency of the process? Do they have reference literature, or have they carried out investigations to assess the reliability of the procedure? Please, discuss the topic.

lines 354-414: While obtaining primary cultures of this type is valuable, if the cells are derived from patients with calcific aortic valve disease or coronary artery disease, could their intrinsic predisposition not subtly influence the response to PA compared with OA, even when using the BSA-treated control?

line 408: Are there any plots available to address this point?

lines 479-482: As previously recommended, it would be useful to generate graphs with GraphPad, organized by category, also for the qPCR data.

lines 489-496: This part is not pertinent to the conclusions and should be moved to the Discussion section. The Conclusion should therefore be rewritten accordingly.

In conclusion, the manuscript in its current form requires  a major revision.

Author Response

We sincerely thank the reviewer for the constructive criticism and valuable suggestions, which significantly helped us to improve the manuscript. We provide a point-by-point response to the reviewer’s suggestions in the attachment.

Reviewer 2 Report

Comments and Suggestions for Authors

This manuscript investigates the effects of palmitic acid (PA) and oleic acid (OA) on endothelial cells derived from different vascular beds. The subject is timely and relevant, given the central role of dietary lipids in the development of vascular inflammation and cardiovascular disease. The use of multiple endothelial cell types is a strength of the work, and the experiments are competently performed. However, in its current form, the study remains descriptive mainly, and several important issues need to be addressed.

The novelty of the work is somewhat overstated. The detrimental effects of PA on endothelial cells, in contrast to the neutral or protective effects of OA, are already well-documented. The main new element here is the side-by-side comparison across different endothelial cell types, which, while potentially useful, represents only a modest advance. The introduction should better acknowledge the extensive prior literature and provide a more careful definition of what this study contributes.

Mechanistic depth is lacking. The study shows that PA induces inflammatory markers, oxidative stress, and impaired function, whereas OA does not, but the results stop at association. Without functional experiments, such as the inhibition of NF-κB, TLR4, or ER stress pathways, or rescue approaches, it is not possible to establish causality. The discussion should be revised to acknowledge this limitation and avoid language that implies mechanistic insight.

The inclusion of endothelial cells from multiple vascular beds is a valuable feature, but the differences between cell types are not analyzed in sufficient depth. A more systematic comparison of shared versus distinct responses would strengthen the manuscript. Currently, the aspect of heterogeneity appears underdeveloped.

The discussion tends to overstate the implications of the work, suggesting that the results provide a mechanistic framework for lipid-driven vascular inflammation. In reality, the data confirm established effects of PA and OA rather than extending them into new mechanistic territory. Clinical extrapolations, particularly those related to diet, should be approached with caution, as this study is limited to in vitro models.

Finally, the reference list needs to be updated. Several recent studies on fatty acid-induced endothelial dysfunction, ER stress, and metabolic inflammation are missing and should be cited to place the findings in the context of current knowledge.

In summary, the manuscript presents an interesting dataset that highlights the differential effects of PA and OA on endothelial cells from various vascular origins. However, the work remains descriptive and incremental, and ideally, mechanistic validation would be necessary to elevate the study's impact.

Comments on the Quality of English Language

The writing is understandable but repetitive, with exaggerated phrasing that should be tightened.

Author Response

(The authors gave the same response as above.)

Round 2

Reviewer 2 Report

Comments and Suggestions for Authors

The revised version of the manuscript exhibits several notable improvements in organization, clarity, and the citation of relevant literature. The inclusion of more recent studies and a more thorough acknowledgment of prior work strengthen the background section. The language is clear and the figures are more readable than in the previous version. However, the revision does not fully resolve the key scientific weaknesses that limited the original submission.

The main limitation remains that the study is descriptive rather than mechanistic. While the experiments demonstrate that palmitic acid (PA) induces inflammation, oxidative stress, and endothelial dysfunction, whereas oleic acid (OA) does not, these effects are already well established in the literature. The novelty of comparing multiple endothelial cell types is appreciated, but this alone represents only an incremental advance. The authors have not performed mechanistic experiments (e.g., inhibition or rescue studies involving NF-κB, TLR4, ER stress, or antioxidant pathways) that would allow causal conclusions to be drawn. Without such data, the work cannot move beyond confirming previously known phenomena.

The comparative aspect of endothelial heterogeneity, which was a potential strength of the study, remains underdeveloped. The manuscript mentions differences among vascular beds but does not provide a systematic analysis of cell-type–specific responses or common versus divergent pathways. A more analytical approach here would have strengthened the paper’s impact.

The discussion has been improved in tone, but still occasionally overstates the mechanistic and translational implications of the findings. Phrases suggesting that this work “provides a mechanistic framework” or implies direct dietary or clinical relevance should be revised to reflect the descriptive and in vitro nature of the data.

In summary, the authors have clearly made an effort to refine the manuscript, but the core scientific limitations remain unresolved. The study confirms the known pro-inflammatory effects of PA and the neutral behavior of OA across endothelial cell types, but does not provide new mechanistic insight. To enhance the impact and originality of the work, it should incorporate functional or interventional experiments and a more comprehensive comparative analysis of endothelial heterogeneity.

Author Response

We sincerely thank the reviewer for the constructive criticism and valuable suggestions, which significantly helped us to improve the manuscript. In the attached file, we provide a point-by-point response to the reviewer’s suggestions. Please see the attachment.
